# DiffIR2VR-Zero: Zero-Shot Video Restoration with Diffusion-based Image Restoration Models

## Abstract

This paper introduces a method for zero-shot video restoration using pre-trained image restoration diffusion models. Traditional video restoration methods often need retraining for different settings and struggle with limited generalization across various degradation types and datasets. Our approach uses a hierarchical latent warping strategy for keyframes and local frames, combined with token merging that uses a hybrid correspondence mechanism that integrates spatial information, optical flow, and feature-based matching. We show that our method not only achieves top performance in zero-shot video restoration but also significantly surpasses trained models in generalization across diverse datasets and extreme degradations ($8\times$ super-resolution and high-standard deviation video denoising). We present evidence through quantitative metrics and visual comparisons on various challenging datasets. Additionally, our technique works with any 2D restoration diffusion model, offering a versatile and powerful tool for video enhancement tasks without extensive retraining.

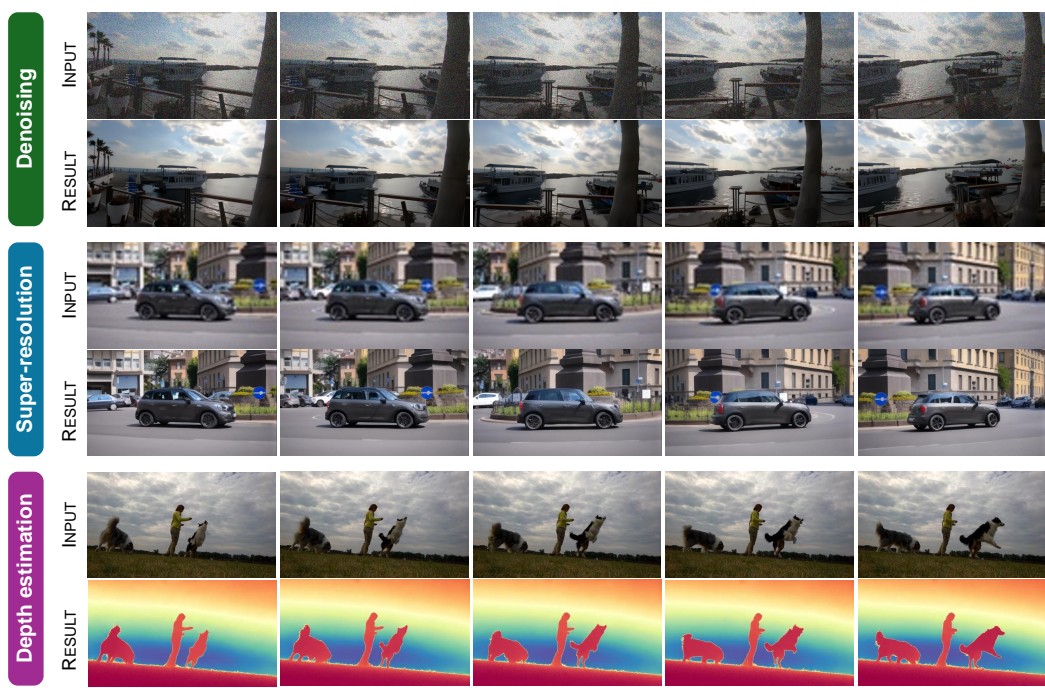

Figure 1: **Zero-shot temporal-consistent diffusion model for video restoration and beyond.** Given a pre-trained diffusion model for *single-image* restoration, our method generates temporally consistent restored video with fine details *without* any further training. Our method applies to other video applications, such as depth estimation.

# 1 INTRODUCTION

Diffusion models have recently achieved remarkable success in image restoration tasks (Xia et al., 2023; Lin et al., 2024). These models can generate realistic details, overcoming the limitations of traditional regression-based methods that often produce blurry outputs without fine details (Fig. 2 (a)). The state-of-the-art methods employing convolutional neural networks (CNNs) (Albawi et al., 2017; Kalchbrenner et al., 2014; O'shea & Nash, 2015) or transformers (Dosovitskiy et al., 2021; Liu et al., 2021b; Vaswani et al., 2017) trained on large-scale data have shown incredible effectiveness in image restoration.

Given the success of diffusion models in image restoration, a natural extension is to apply them to video restoration. Video restoration, which typically involves denoising, super-resolution, and deblurring, is a valuable field that transforms low-quality videos into high-quality ones. However, directly applying image-based diffusion models to video restoration presents significant challenges. Notably, performing per-frame inference on videos using these models often results in severe flickering (Fig. 2 (b)), especially when using Latent Diffusion Models (LDMs).

Surprisingly, the application of image restoration diffusion models to video restoration remains largely unexplored. While some attempts have been made to adapt these models for video tasks, they typically involve fine-tuning with 3D convolution and temporal attention layers. However, such approaches require extensive computational resources (*e.g.*, 32 A100-80G GPUs for video upscaling (Zhou et al., 2023)) and task-specific retraining, limiting their practicality and generalizability.

In this paper, we present a novel, training-free approach to leverage pre-trained image restoration diffusion models for video restoration. Our method introduces two key modules: hierarchical latent warping and hybrid flow-guided spatial-aware token merging. These modules work in tandem to enforce temporal consistency in both latent and token (feature from the attention layer) spaces, enabling high-quality video restoration without any additional training or fine-tuning. Our method (Fig. 2 (c)) achieves both realistic and temporally consistent results without any additional training, leveraging an image diffusion model to restore videos effectively.

Fig. 1 illustrates our method's capability to generate temporally consistent restored videos across various tasks, including denoising, super-resolution, and depth estimation, *without* any further training. Our zero-shot video restoration framework can be applied to any pre-trained image diffusion model, offering a versatile solution that can adapt to various restoration tasks.

While inspired by recent advances in diffusion-based generation models like VidToMe (Li et al., 2024) and TokenFlow (Geyer et al., 2023), our work goes beyond combining existing techniques. Our main contributions are:

- First zero-shot video restoration using diffusion models, balancing temporal consistency and detail generation across various image-based models.
- Training-free framework manipulating latent and token spaces with hierarchical latent warping and improved token merging.
- State-of-the-art results in extreme scenarios, surpassing traditional methods in generalizability and robustness.

# 2 RELATED WORK

**Video Restoration.** Video restoration aims to restore high-quality frames from degraded videos, addressing issues such as noise, blur, and low resolution (Chan et al., 2021a;c; Isobe et al., 2020; Li et al., 2023; Youk et al., 2024; Zhang et al., 2018; Liu et al., 2019; 2021a). This task is more challenging than image restoration (Guo et al., 2019) due to the need for temporal consistency. Learning-based approaches employ architectures like optical flow warping (Huang et al., 2022; Pan et al., 2020; Shi et al., 2023a;b; Xue et al., 2019), deformable convolutions (Chan et al., 2021a;b; Dai et al., 2017; Tian et al., 2020; Wang et al., 2019; 2020; Zhu et al., 2019), and attention mechanisms to handle temporal dependencies (Cao et al., 2021; Li et al., 2020; Liang et al., 2022; Zamir et al., 2022). Major limitations include dependency on paired HQ-LQ data (Chan et al., 2022b; Xie et al., 2023; Yang et al., 2021), assumptions of predefined degradation processes (Kim et al., 2017; 2016; Kong et al., 2023; Li et al., 2020; Liang et al., 2022), and the need for retraining for different degradation levels (Liu & Sun, 2013; Nah et al., 2019; Yi et al., 2019; Youk et al., 2024). These factors reduce effectiveness in real-world applications and lead to poor generalization. Additionally, these methods

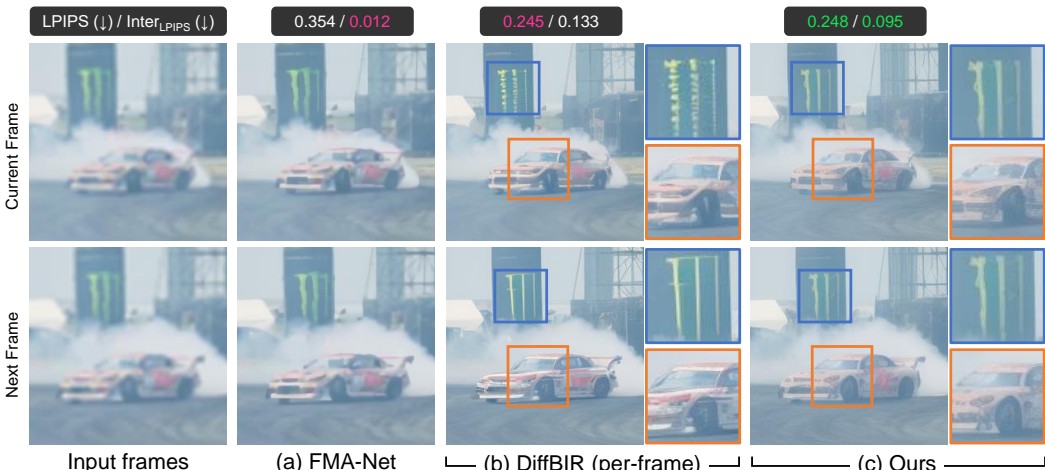

Figure 2: **4× video super-resolution results.** (a) Traditional regression-based methods such as FMA-Net (Youk et al., 2024) are limited to the training data domain and tend to produce blurry results when encountering out-of-domain inputs. (b) Although applying image-based diffusion models such as DiffBIR (Lin et al., 2024) to individual frames can generate realistic details, these details often lack consistency across frames. (c) Our method leverages an image diffusion model to restore videos, achieving both realistic and consistent results *without* any additional training.

often lose significant detail, similar to image restoration (Chen et al., 2022; Liang et al., 2021; Wang et al., 2021; Zhang et al., 2021).

**Diffusion Models for Image Restoration.**  With significant advancements in diffusion models (Choi et al., 2021; Dhariwal & Nichol, 2021; Hertz et al., 2023; Ho et al., 2020; Rombach et al., 2022), many diffusion-based approaches have been proposed for image restoration (Fei et al., 2023; Ho et al., 2020; Nichol et al., 2021; Sohl-Dickstein et al., 2015; Song et al., 2020b; Wang et al., 2023; Yang et al., 2023b). These methods include training diffusion models from scratch (Rombach et al., 2022; Saharia et al., 2022; Xia et al., 2023; Yue et al., 2024), introducing constraints into the reverse diffusion process of pre-trained models (Kawar et al., 2022), and fine-tuning frozen pre-trained diffusion models with additional trainable layers (Wang et al., 2023; Yang et al., 2023b; Zhang et al., 2023), as seen in StableSR (Wang et al., 2023) and DiffBIR (Lin et al., 2024). Despite their effectiveness in image restoration, these methods face challenges in video restoration due to temporal inconsistencies caused by the diffusion process's randomness. In contrast, our method allows these approaches to work on video without any training, addressing the temporal consistency issue while leveraging the strengths of image restoration diffusion models.

**Video Editing Methods for Temporal Consistency.**  Recent research has extended pre-trained image diffusion models to video tasks (Esser et al., 2023; Ho et al., 2022a;b; Hu et al., 2023; Lu et al., 2023; Luo et al., 2023; Mei & Patel, 2023; Kara et al., 2024). Various methods have been proposed to enhance temporal consistency in video editing, which can be categorized based on the level at which they operate:

- Latent Space Level: Approaches working at the latent space level include Rerender-A-Video (Yang et al., 2023a), which employs latent warping (Teed & Deng, 2020; Xu et al., 2022) and frame interpolation. While these methods aim to maintain consistency in the latent representations of consecutive frames, they may struggle with semantic consistency in demanding restoration tasks. Our method introduces a novel hierarchical latent warping technique that addresses these limitations.

- Token Level: Methods operating at the token level include VidToMe (Li et al., 2024) and TokenFlow (Geyer et al., 2023), which enhance temporal consistency by merging attention tokens across frames. Token merging (Bolya et al., 2023) is another technique used at this level. However, these techniques often produce blurry outputs in restoration tasks. Our approach improves upon these methods by introducing a hybrid flow-guided spatial-aware token merging technique that maintains sharpness while ensuring temporal consistency.

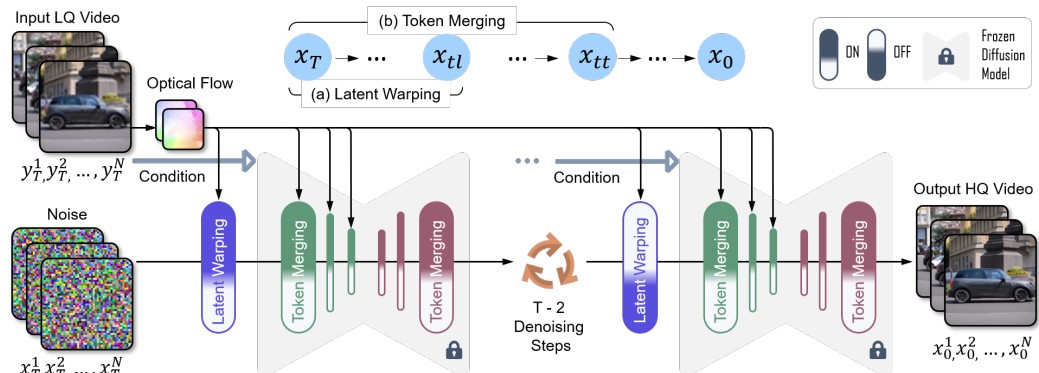

Figure 3: **Pipeline of our proposed zero-shot video restoration method.** We process low-quality (LQ) videos in batches using a diffusion model, with a keyframe randomly sampled within each batch. (a) At the beginning of the diffusion denoising process, hierarchical latent warping provides rough shape guidance both globally, through latent warping between keyframes, and locally, by propagating these latents within the batch. (b) Throughout most of the denoising process, tokens are merged before the self-attention layer. For the downsample blocks, optical flow is used to find the correspondence between tokens, and for the upsample blocks, cosine similarity is utilized. This hybrid flow-guided, spatial-aware token merging accurately identifies correspondences between tokens by leveraging both flow and spatial information, thereby enhancing overall consistency at the token level.

While these video editing techniques generate impressive results with minimal effort, they often struggle with semantic consistency and detail preservation in demanding restoration tasks. Our work draws inspiration from these approaches but introduces novel elements specifically designed to address the challenges of video restoration, combining the strengths of latent and token-level methods while mitigating their individual weaknesses.

## 3 METHOD

Given a low-quality video with $n$ frames $\{y^1, y^2, \ldots, y^n\}$, we aim to restore it to high-quality $\{x^1, x^2, \ldots, x^n\}$ using image-based diffusion models. Directly applying these models frame-by-frame causes temporal inconsistency due to inherent stochasticity, especially in extreme degradation (Fig. 2 and Fig. 6). Our method (Fig. 3) addresses this by enforcing temporal stability in latent and token spaces through Hierarchical Latent Warping (Sec. 3.2) and Hybrid Flow-guided Spatial-aware Token Merging (Sec. 3.3). We introduce diffusion models and video token merging (Sec. 3.1), then detail our key components (Sec. 3.2-Sec. 3.4).

### 3.1 DIFFUSION MODELS FOR VIDEO EDITING

Diffusion models have been successfully applied to video editing tasks by extending image-based models. These models typically operate as follows:

**Diffusion Process.** The forward process adds noise to a clean image $x_0$ over $T$ steps:

$$x_t = \sqrt{\alpha_t}x_{t-1} + \sqrt{1-\alpha_t}\epsilon_{t-1} \Rightarrow x_t = \sqrt{\bar{\alpha}_t}x_0 + \sqrt{1-\bar{\alpha}_t}\epsilon, \qquad (1)$$

where $t \sim [1, T]$, $\epsilon_t, \epsilon \sim \mathcal{N}(\mathbf{0}, \mathbf{I})$, and $\bar{\alpha}_t = \prod_{s=1}^{t}\alpha_s$. A UNet-based denoiser $\epsilon_\theta$ is trained to estimate and remove this noise. During inference, the inverse process gradually denoises $x_t$ to produce $x_0$ (Ho et al., 2020; Song et al., 2020a; 2023). These models can be enhanced with additional guidance signals for controlled generation (Zhang et al., 2023; Kawar et al., 2022).

**Video Token Merging.** To maintain temporal consistency, techniques like Video Token Merging (VidToMe) (Li et al., 2024) are employed. This process merges similar tokens within frame chunks in attention blocks: Given a token chunk $\mathbf{T} \in \mathbb{R}^{B \times A \times C}$, where $A = w * h$, the algorithm first separates the tokens into source tokens $\mathbf{T}_{\mathrm{src}} \in \mathbb{R}^{B \times A-1 \times C}$ and a target token $\mathbf{T}_{\mathrm{tar}} \in \mathbb{R}^{B \times 1 \times C}$. It then calculates the cosine between each source and target token, determining their corresponding similarity levels, denoted $score \in \mathbb{R}^{((B-1)*A) \times A}$. The algorithm then identifies the most similar target token for each source token by taking the maximum value in the last column.

$$s(\mathbf{T}_{\mathrm{src}}, \mathbf{T}_{\mathrm{tar}}) = \frac{\mathbf{T}_{\mathrm{src}} \cdot \mathbf{T}_{\mathrm{tar}}}{\|\mathbf{T}_{\mathrm{src}}\| \, \|\mathbf{T}_{\mathrm{tar}}\|} \, , \quad c = \max_{\{\mathbf{t} \in \mathbf{T}_{\mathrm{tar}}\}}(s(\mathbf{T}_{\mathrm{src}}, \mathbf{t})), \qquad (2)$$

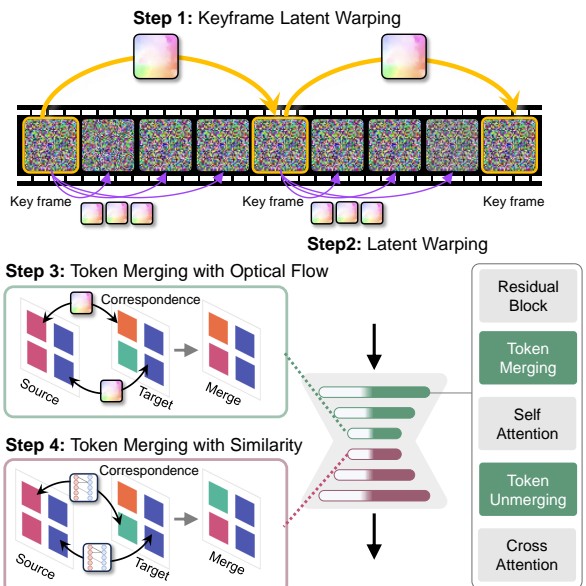

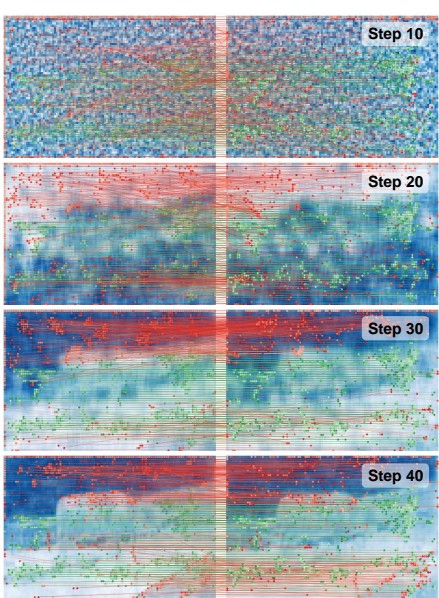

Figure 4: **An illustration of our key modules.** Without requiring any training, these modules can achieve coherence across frames by enforcing temporal stability in both latent and token space. Hierarchical latent warping provides global and local shape guidance; Hybrid spatial-aware token merging before the self-attention layer improves temporal consistency by matching similar tokens using optical flow in the down blocks and cosine similarity in the up blocks of the UNet.

Figure 5: **Token correspondences (cosine similarity and optical flow) across denoising steps.** Early on (e.g. step 10), optical flow guides better due to noisy latents. Later (*e.g.* steps 30-40), similarity and flow focus on different regions, showcasing the benefit of our hybrid approach for effective token merging throughout denoising.

where $s(\cdot, \cdot)$ is the cosine similarity score and $c$ indicates the correspondences. Next, the $r$ most similar paired source-target tokens are merged, and the remaining tokens are concatenated as the output. Merged tokens are subsequently unmerged after self-attention to preserve the original shape by simply assigning the merged source-target tokens the exact same value. The token merging and unmerging are defined as follows:

$$\mathbf{T}_{\text{merge}} = \mathcal{M}(\mathbf{T}_{\text{src}}, \mathbf{T}_{\text{tar}}, \ c, \ r), \quad \mathbf{T}_{\text{unmerge}} = \mathcal{U}(\mathbf{T}_{\text{merge}}, \ c), \tag{3}$$

where $\mathcal{M}$ and $\mathcal{U}$ denote the merging and unmerging operations, respectively.

**Latent Warping.** Some methods (Zhou et al., 2023) perform warping in the latent space to maintain consistency between frames. This is done by warping the latent representations of adjacent frames.

**Limitations in video restoration.** Existing video editing techniques face challenges in video restoration, often prioritizing temporal consistency over detail preservation. Early-stage denoising produces noisy latents, making traditional similarity measures unreliable, especially in UNet's downsample blocks (Fig. 5, top). Most methods focus on frame-to-frame consistency and missing global-local coherence, while high merging ratios can lead to over-smoothing. Our approach combines hierarchical latent warping with hybrid flow-guided spatial-aware token merging to address these limitations. This combination provides multi-scale temporal consistency, balances detail preservation with consistency, and adapts to various degradation types. Latent warping handles large-scale inconsistencies in early stages, while token merging ensures fine detail consistency as features become more meaningful. By leveraging both optical flow and similarity measures, our method aims for superior zero-shot video restoration without task-specific training or computational resources.

## 3.2 HIERARCHICAL LATENT WARPING

We introduce a hierarchical latent warping module operating in latent space, with a two-level approach: (1) Global level: Warping between keyframes, and (2) Local level: Propagating warped latents within each batch. As shown in Fig. 4 (upper part), this provides rough shape guidance on global and local

scales. Let $\hat{x}_{t\to0}^i$ be the predicted $\hat{x}_0$ latent for the $i^{th}$ keyframe at denoising step $t$. We first perform global-level warping between keyframes:

$$\hat{x}_{t\to0}^i \leftarrow M_{ji} \cdot \hat{x}_{t\to0}^i + (1 - M_{ji}) \cdot \mathcal{W}(\hat{x}_{t\to0}^j, f_{ji}), \tag{4}$$

where $j = i-1$ and $f_{ji}$, $M_{ji}$ denotes the optical flow and the occlusion mask from $lq_j$ to $lq_i$ estimated by GMFlow (Xu et al., 2022). We then perform local-level warping by propagating these latents to remaining frames within each batch. This approach ensures corresponding points share similar latents globally across keyframes and locally within batches from the start of denoising, providing a more comprehensive approach to maintaining consistency compared to simple frame-to-frame warping.

### 3.3 HYBRID FLOW-GUIDED SPATIAL-AWARE TOKEN MERGING

While latent manipulation can achieve a certain degree of consistency, manipulating latents during the later stages of the denoising process would result in blurry outcomes. Additionally, the token space is highly semantically related to the image. Therefore, we propose hybrid flow-guided spatial-aware token merging to achieve consistency in the token space.

**Flow-guided.**  Our hybrid correspondence mechanism integrates spatial information, optical flow, and feature-based similarity. In early denoising stages, latents are noisy, making cosine similarity unreliable, especially in UNet's downsample blocks (Fig. 5, top). However, optical flow from low-resolution inputs provides better guidance. As denoising progresses (*e.g.*, steps 30-40), flow-based and similarity-based methods often identify different matches (Fig. 5, bottom), suggesting the benefit of a hybrid approach. Even with low-quality video, we can identify correspondences between frames based on color. We use flow for correspondences in UNet downsample blocks and employ forward-backward consistency check as a criterion to determine $r$ most similar paired source token $\mathbf{T}_{src}$ and target token $\mathbf{T}_{tar}$:

$$\sigma = \exp(- \|f_{src\to tar}(X(\mathbf{T}_{src})) + f_{tar\to src}(X(\mathbf{T}_{src}) + f_{src\to tar}(X(\mathbf{T}_{src})))\|_2^2), \tag{5}$$

where $\sigma$ is the confidence, $X(\mathbf{T}_{src})$ is the spatial location of $\mathbf{T}_{src}$, and $f_{src\to tar}$, $f_{tar\to src}$ denotes the forward and backward flow between $\mathbf{T}_{src}$ and $\mathbf{T}_{tar}$. The proposed flow-guided token merging is:

$$\mathbf{T}_{merge} = \mathcal{M}(\mathbf{T}_{src}, \mathbf{T}_{tar}, f_{src\to tar}, \sigma, r). \tag{6}$$

Fig. 4 provides a clearer illustration of our proposed component. While optical flow can be challenging in certain conditions (*e.g.*, fast motion, textureless regions), our method incorporates several safeguards. We use forward-backward consistency checks, merge only the $r$ most similar token pairs, and combine flow-based correspondence with spatial information and similarity matching. This multi-faceted approach ensures robust performance in challenging conditions. Additionally, as shown in Fig. 5 (bottom), flow and cosine similarity identify different correspondences, providing comprehensive guidance. Tab. 1 demonstrates that using flow correspondence in downblocks and similarity in upblocks yields the best visual quality and temporal consistency.

**Spatial-awareness and Padding Removal.**  Directly finding correspondences using cosine similarity can lead to mismatches in areas with uniform textures, especially in video backgrounds (*e.g.*, sky, sand, grass; Fig. 5, bottom), resulting in blurrier outcomes. Given that corresponding points in adjacent frames are typically spatially close, we leverage this information by weighting cosine similarity scores with tokens' spatial distances:

$$s'_{ij} = s_{ij} \cdot e^{-\tau}, \text{ with } \tau = \left\lfloor \left[ \|X(i) - X(j)\|_2^2 \right] / R \right\rfloor, \tag{7}$$

where $X(i)$, $X(j)$ are spatial locations of the $i^{th}$ source and $j^{th}$ target token; $R$ is a hyperparameter defining the radius of the uniform weight region.

This spatial awareness primarily applies to cosine similarity correspondences in UNet upsample blocks. For flow correspondences in downsample blocks, we rely on forward-backward consistency checks as described in Eq. (5), since optical flow models inherently consider spatial information. This combination ensures effective utilization of spatial information throughout our token merging process. Another point to consider is that images are often padded to pass through the UNet, which can significantly impact token correspondences by causing cosine similarity to mistakenly align padding with actual content, even in later denoising stages. To mitigate this, we remove padding before merging and reapply it after unmerging. See the appendix for visual ablation results.

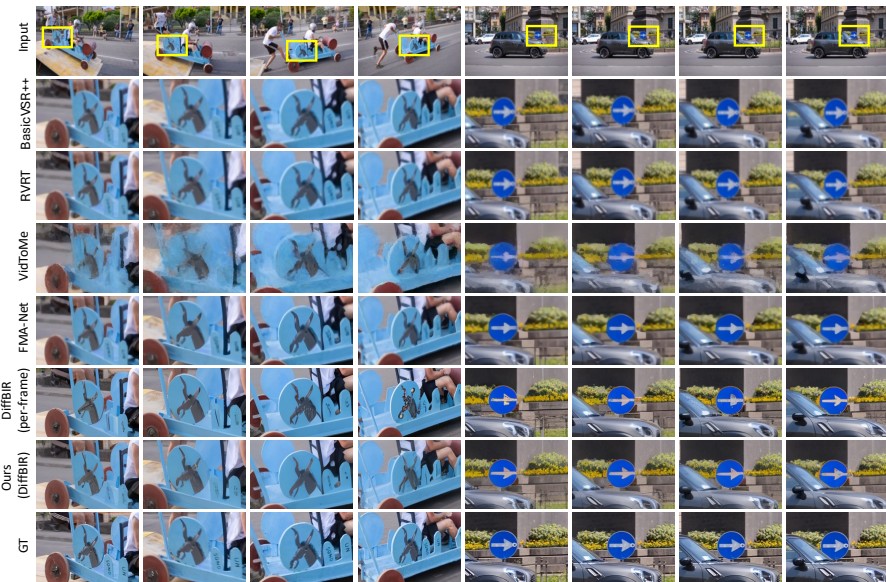

Figure 6: **Qualitative comparisons on 4× video super-resolution.** As shown in the first row, the low-quality input lacks almost all details. In the zoomed-in patches, our method produces clearer and more consistent results.

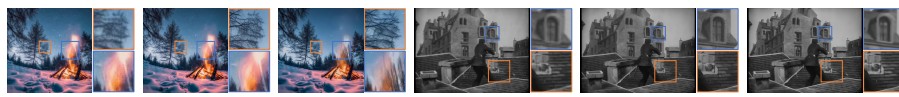

Figure 7: **Qualitative comparisons with Upscale-A-Video (Zhou et al., 2023) on 4× video SR.**

**Token Unmerging.** After the self-attention operation, tokens need to be unmerged to restore the original shape. We adopt a replacement-based unmerging process where tokens are restored to their original shape using the identified correspondences. This approach is similar to VidToMe (Li et al., 2024), but our method's primary innovation lies in enhancing the correspondence identification process during the merging stage, which leads to more accurate and effective token matching.

**Merging Ratio Annealing.** To prevent over-smoothing in later denoising stages, we employ ratio annealing to gradually reduce the merging ratio. The merging ratio of the $i^{th}$ denoising step is:

$$r_i = r \cdot \cos\left(\frac{\pi}{2} \cdot \max\left(\min\left(\delta \cdot \frac{i - i_{\text{beg}}}{i_{\text{end}} - i_{\text{beg}}}, 1\right), 0\right)\right), \tag{8}$$

where $i_{\text{beg}}$, $i_{\text{end}}$ are predefined steps indicating the beginning and end of the merging process, and $\delta$ controls annealing speed. This technique balances smoothness and temporal consistency, achieving a compromise between regression-based methods (temporally consistent but overly smooth) and per-frame inferencing (detailed but inconsistent). As shown in Fig. 2 and Fig. 6, our approach preserves fine details while maintaining temporal coherence, proving effective in severe degradation scenarios. Visual comparisons for 8× super-resolution are provided in supplementary materials.

### 3.4 SCHEDULING

As depicted in Fig. 3, at the initial stage of the diffusion denoising process, hierarchical latent warping offers rough shape guidance on a global scale by warping latents between keyframes and on a local scale by propagating these latents within the batch. During the majority of the denoising process, tokens are processed with our hybrid spatial-aware token merging before entering the attention layer. This component further improves temporal consistency by matching similar tokens, utilizing both flow and spatial information.

## 4 EXPERIMENTS

**Testing Dataset.** For video super-resolution, we evaluate on REDS4 (Nah et al., 2019), Vid4 (Liu & Sun, 2013) and DAVIS (Perazzi et al., 2016a) testing sets, with downsample scales ×4 and ×8,

Table 1: **Quantitative comparisons.** (*Left*) Video super-resolution on the DAVIS (Perazzi et al., 2016b), Vid4 (Liu & Sun, 2013) and REDS4 (Nah et al., 2019) datasets. (*Right*) video denoising of various noise levels on the REDS30 and Set8 (Tassano et al., 2019) dataset. The best and second performances are marked in red and blue, respectively. $E^*_{\text{warp}}$ denotes $E_{\text{warp}}(\times 10^{-3})$ and $E_{\text{inter}}$, LPIPS$_{\text{inter}}$ denotes interpolation error and LPIPS. - indicates out-of-memory.

| | Metrics | VidToMe | FMA-Net | SD ×4 Frame | SD ×4 Ours | DiffBIR Frame | DiffBIR Ours |
|---|---|---|---|---|---|---|---|
| DAVIS ×4 | PSNR ↑ | 23.014 | 25.215 | 23.504 | 23.843 | 23.780 | 24.182 |
| | SSIM ↑ | 0.566 | 0.727 | 0.584 | 0.618 | 0.601 | 0.621 |
| | LPIPS ↓ | 0.405 | 0.347 | 0.277 | 0.272 | 0.264 | 0.262 |
| | $E^*_{\text{warp}}$ ↓ | 0.520 | 0.186 | 0.912 | 0.745 | 0.654 | 0.474 |
| | $E_{\text{inter}}$ ↓ | 13.676 | 11.558 | 18.125 | 17.431 | 16.529 | 14.666 |
| | LPIPS$_{\text{inter}}$ ↓ | 0.329 | 0.078 | 0.292 | 0.274 | 0.266 | 0.232 |
| DAVIS ×8 | PSNR ↑ | 22.097 | 22.690 | 20.268 | 20.519 | 21.964 | 22.331 |
| | SSIM ↑ | 0.513 | 0.594 | 0.446 | 0.424 | 0.502 | 0.519 |
| | LPIPS ↓ | 0.554 | 0.528 | 0.470 | 0.434 | 0.362 | 0.367 |
| | $E^*_{\text{warp}}$ ↓ | 0.440 | 0.351 | 2.199 | 1.759 | 0.964 | 0.699 |
| | $E_{\text{inter}}$ ↓ | 12.624 | 13.978 | 24.496 | 21.746 | 17.981 | 15.853 |
| | LPIPS$_{\text{inter}}$ ↓ | 0.388 | 0.132 | 0.457 | 0.442 | 0.372 | 0.333 |
| REDS4 ×4 | PSNR ↑ | 23.134 | 25.829 | 24.189 | 24.226 | 24.679 | 25.118 |
| | SSIM ↑ | 0.589 | 0.761 | 0.638 | 0.641 | 0.657 | 0.683 |
| | LPIPS ↓ | 0.357 | 0.327 | 0.247 | 0.242 | 0.211 | 0.222 |
| | $E^*_{\text{warp}}$ ↓ | 0.579 | 0.392 | 0.817 | 0.811 | 0.704 | 0.502 |
| | $E_{\text{inter}}$ ↓ | 17.869 | 19.014 | 22.906 | 22.889 | 22.305 | 20.130 |
| | LPIPS$_{\text{inter}}$ ↓ | 0.356 | 0.133 | 0.295 | 0.281 | 0.271 | 0.221 |
| REDS4 ×8 | PSNR ↑ | 21.894 | 22.842 | - | - | 22.479 | 22.961 |
| | SSIM ↑ | 0.532 | 0.644 | - | - | 0.559 | 0.59 |
| | LPIPS ↓ | 0.538 | 0.423 | - | - | 0.311 | 0.306 |
| | $E^*_{\text{warp}}$ ↓ | 0.423 | 0.753 | - | - | 0.828 | 0.551 |
| | $E_{\text{inter}}$ ↓ | 15.502 | 21.519 | - | - | 21.76 | 19.382 |
| | LPIPS$_{\text{inter}}$ ↓ | 0.412 | 0.159 | - | - | 0.351 | 0.287 |
| REDS4 ×16 | PSNR ↑ | 20.520 | 21.569 | 18.706 | 18.858 | 20.124 | 20.712 |
| | SSIM ↑ | 0.483 | 0.570 | 0.461 | 0.410 | 0.461 | 0.509 |
| | LPIPS ↓ | 0.697 | 0.565 | 0.612 | 0.562 | 0.446 | 0.438 |
| | $E^*_{\text{warp}}$ ↓ | 0.296 | 0.619 | 2.664 | 2.030 | 1.168 | 0.665 |
| | $E_{\text{inter}}$ ↓ | 12.945 | 18.758 | 28.478 | 24.000 | 21.33 | 17.731 |
| | LPIPS$_{\text{inter}}$ ↓ | 0.417 | 0.139 | 0.559 | 0.493 | 0.444 | 0.358 |
| Vid4 ×4 | PSNR ↑ | 19.622 | 23.209 | 20.047 | 20.134 | 20.687 | 21.226 |
| | SSIM ↑ | 0.425 | 0.679 | 0.478 | 0.473 | 0.497 | 0.525 |
| | LPIPS ↓ | 0.491 | 0.375 | 0.343 | 0.331 | 0.329 | 0.326 |
| | $E^*_{\text{warp}}$ ↓ | 0.687 | 0.203 | 1.502 | 1.397 | 1.156 | 0.677 |
| | $E_{\text{inter}}$ ↓ | 11.754 | 4.442 | 17.234 | 16.921 | 15.478 | 11.316 |
| | LPIPS$_{\text{inter}}$ ↓ | 0.337 | 0.026 | 0.275 | 0.271 | 0.265 | 0.198 |
| Vid4 ×8 | PSNR ↑ | 18.811 | 21.033 | 17.813 | 17.992 | 18.636 | 19.304 |
| | SSIM ↑ | 0.372 | 0.521 | 0.345 | 0.307 | 0.367 | 0.406 |
| | LPIPS ↓ | 0.654 | 0.514 | 0.507 | 0.484 | 0.440 | 0.435 |
| | $E^*_{\text{warp}}$ ↓ | 0.477 | 0.221 | 2.523 | 1.972 | 1.524 | 0.767 |
| | $E_{\text{inter}}$ ↓ | 9.942 | 5.269 | 22.881 | 19.970 | 18.112 | 12.281 |
| | LPIPS$_{\text{inter}}$ ↓ | 0.393 | 0.032 | 0.423 | 0.419 | 0.395 | 0.294 |

| $\sigma$ | Metrics | VidToMe | Shift-Net | DiffBIR Frame | DiffBIR Ours |
|---|---|---|---|---|---|
| REDS30 75 | PSNR ↑ | 22.671 | 21.033 | 24.585 | 24.520 |
| | SSIM ↑ | 0.559 | 0.381 | 0.649 | 0.649 |
| | LPIPS ↓ | 0.397 | 0.735 | 0.276 | 0.275 |
| | $E^*_{\text{warp}}$ ↓ | 0.727 | 0.765 | 0.751 | 0.706 |
| | $E_{\text{inter}}$ ↓ | 18.440 | 21.751 | 21.798 | 21.166 |
| | LPIPS$_{\text{inter}}$ ↓ | 0.375 | 0.501 | 0.275 | 0.264 |
| REDS30 100 | PSNR ↑ | 22.588 | 22.573 | 24.524 | 24.534 |
| | SSIM ↑ | 0.557 | 0.484 | 0.648 | 0.652 |
| | LPIPS ↓ | 0.404 | 0.518 | 0.275 | 0.271 |
| | $E^*_{\text{warp}}$ ↓ | 0.733 | 1.126 | 0.763 | 0.696 |
| | $E_{\text{inter}}$ ↓ | 18.370 | 23.424 | 21.835 | 20.639 |
| | LPIPS$_{\text{inter}}$ ↓ | 0.380 | 0.375 | 0.281 | 0.267 |
| REDS30 random | PSNR ↑ | 22.348 | 21.113 | 24.579 | 24.508 |
| | SSIM ↑ | 0.546 | 0.386 | 0.650 | 0.649 |
| | LPIPS ↓ | 0.429 | 0.728 | 0.276 | 0.270 |
| | $E^*_{\text{warp}}$ ↓ | 0.681 | 1.896 | 0.755 | 0.713 |
| | $E_{\text{inter}}$ ↓ | 17.608 | 27.565 | 21.743 | 21.140 |
| | LPIPS$_{\text{inter}}$ ↓ | 0.384 | 0.542 | 0.282 | 0.272 |
| Set8 50 | PSNR ↑ | 21.531 | 23.433 | 23.197 | 23.713 |
| | SSIM ↑ | 0.501 | 0.482 | 0.594 | 0.63 |
| | LPIPS ↓ | 0.415 | 0.574 | 0.261 | 0.245 |
| | $E^*_{\text{warp}}$ ↓ | 0.911 | 1.358 | 1.078 | 0.747 |
| | $E_{\text{inter}}$ ↓ | 17.217 | 19.845 | 19.732 | 16.814 |
| | LPIPS$_{\text{inter}}$ ↓ | 0.406 | 0.432 | 0.332 | 0.255 |
| Set8 100 | PSNR ↑ | 21.226 | 18.198 | 22.519 | 22.955 |
| | SSIM ↑ | 0.484 | 0.281 | 0.553 | 0.591 |
| | LPIPS ↓ | 0.472 | 0.733 | 0.338 | 0.323 |
| | $E^*_{\text{warp}}$ ↓ | 0.918 | 2.229 | 1.13 | 0.802 |
| | $E_{\text{inter}}$ ↓ | 17.367 | 24.661 | 20.18 | 17.444 |
| | LPIPS$_{\text{inter}}$ ↓ | 0.421 | 0.619 | 0.372 | 0.286 |
| Set8 150 | PSNR ↑ | 20.209 | 16.136 | 21.005 | 21.418 |
| | SSIM ↑ | 0.443 | 0.291 | 0.486 | 0.544 |
| | LPIPS ↓ | 0.554 | 0.729 | 0.449 | 0.402 |
| | $E^*_{\text{warp}}$ ↓ | 0.972 | 4.279 | 1.207 | 0.832 |
| | $E_{\text{inter}}$ ↓ | 17.872 | 22.343 | 20.729 | 17.616 |
| | LPIPS$_{\text{inter}}$ ↓ | 0.470 | 0.646 | 0.450 | 0.331 |

Table 2: **Ablation studies for 8× VSR on DAVIS (Perazzi et al., 2016a) test sets.** (*Left*) different correspondence matching methods. (*Right*) the proposed components applied at different stages of the denoising process. We apply our two proposed components, hierarchical latent warping (HLW) and hybrid spatial-aware token merging (HS-ToMe), at the early, mid, and late denoising stages.

| Down blocks | Up blocks | Spatial-aware | LPIPS ↓ | $E^*_{\text{warp}}$ ↓ | LPIPS$_{\text{inter}}$ ↓ |
|---|---|---|---|---|---|
| Flow | Flow | – | 0.518 | 1.214 | 0.563 |
| Cos | Cos | – | 0.390 | 0.736 | 0.350 |
| Cos | Flow | – | 0.507 | 1.049 | 0.545 |
| Flow | Cos | – | 0.375 | **0.677** | 0.347 |
| Flow | Cos | ✓ | **0.367** | 0.699 | **0.333** |

| HLW (Sec. 3.2) Early | Mid | Late | HS-ToMe (Sec. 3.3) Early | Mid | Late | LPIPS ↓ | $E^*_{\text{warp}}$ ↓ | LPIPS$_{\text{inter}}$ ↓ |
|---|---|---|---|---|---|---|---|---|
| – | – | – | – | – | – | **0.362** | 0.964 | 0.372 |
| ✓ | – | – | ✓ | – | – | 0.368 | 0.887 | 0.369 |
| ✓ | ✓ | – | ✓ | ✓ | ✓ | 0.43 | 0.804 | 0.383 |
| ✓ | ✓ | ✓ | ✓ | ✓ | ✓ | 0.411 | 0.704 | 0.339 |
| ✓ | – | – | ✓ | ✓ | ✓ | 0.367 | **0.699** | **0.333** |

following the degradation pipeline of RealBasicVSR (Chan et al., 2022b). For video denoising, we evaluate on REDS30 (Nah et al., 2019) and Set8 (Tassano et al., 2020) with different noise levels (std. = 50, 75, 100, 150 and randomly sampled from the range [50, 100]).

**Evaluation Metrics.** We assess (1) image quality via LPIPS, SSIM, and PSNR; (2) temporal consistency, using warping error $E_{\text{warp}}$, interpolation error, and interpolation LPIPS. Since LPIPS better reflects visual quality, we propose interpolation LPIPS, based on the interpolation error used in a previous study (Li et al., 2024), to more accurately measure video continuity from a visual perspective. This involves interpolating a target frame from its previous and next frames and computing the LPIPS between the estimated and target frames.

**Implementation Details.** The experiment is conducted on an NVIDIA RTX 4090 GPU. We apply our method to DiffBIR (Lin et al., 2024) and SDx4 upscaler (sdx, 2023), both image-based diffusion models, to demonstrate the proposed method's compatibility with different models. Note that for

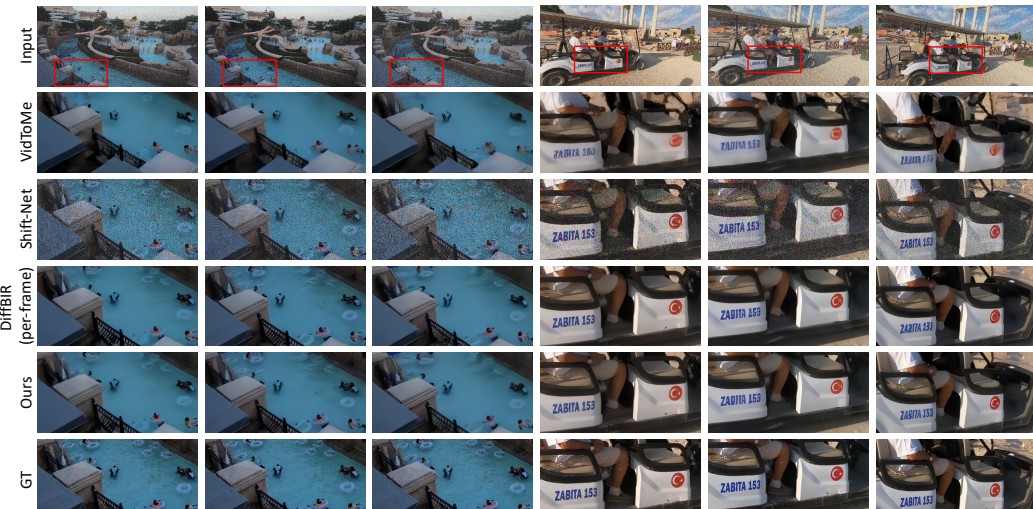

Figure 8: **Video denoising comparisons on the REDS30 (Nah et al., 2019) dataset.** Our method effectively denoises and generates detailed results while maintaining temporal coherence.

models that are restricted to a super-resolution scale of $4\times$, we will apply the process twice and then use bicubic downsampling to achieve $8\times$ results. However, this will can lead to out-of-memory issues for SDx4 upscaler in REDS.

### 4.1 COMPARISONS WITH STATE-OF-THE-ART METHODS

To verify the effectiveness of our approach, we compare it with several state-of-the-art methods, including BasicVSR++ (Chan et al., 2022a), RVRT (Liang et al., 2022), and FMA-Net (Youk et al., 2024) for video super-resolution, and Shift-Net (Li et al., 2023) for video denoising. We also compare our method to per-frame restoration and the application of VidToMe (Li et al., 2024), a zero-shot video editing method, onto the same model as ours. We also try to compare with Upscale-A-Video (Zhou et al., 2023), which is a diffusion-based video restoration model fine-tuned from an image-based diffusion model. However, we are unable to run their inference code on our available hardware (one A6000 GPU, 48GB memory) due to persistent out-of-memory (OOM) issues, even with their default configuration. Therefore, we conduct experiments on the same test cases used in their paper.

Our zero-shot video restoration framework is designed to be highly adaptable and capable of leveraging a wide range of pre-trained image diffusion models. This flexibility allows easy adaptation from image to video models without extensive retraining, enabling the application of various restoration tasks by simply switching the underlying image diffusion model.

**Video Super-resolution.** As shown in Tab. 1, regression-based methods like FMA-Net (Youk et al., 2024) struggle with large motion or severe degradation. VidToMe (Li et al., 2024) can generate highly consistent results, but they are often very blurry, leading to poor visual quality. In contrast, our method enhances temporal consistency while maintaining the generation quality of the original diffusion model, making it the most competitive approach. Fig. 6 provides visualizations of two challenging VSR cases. FMA-Net fails to produce sharp results due to domain gaps between training and testing. Diffusion-based image restoration method DiffBIR (Lin et al., 2024) and SD$\times$4 upscaler (sdx, 2023) can generate sharp results with details, while per-frame processing makes the result video temporal inconsistent and jitters across frames. On the contrary, our zero-shot video restoration framework restores a low-quality input video into a temporally consistent high-quality video. The qualitative comparisons with Upscale-A-Video are provided in Fig. 7. The results demonstrate that our method produces more detailed outputs that better preserve the content of input frames. This advantage stems from our approach of leveraging pre-trained diffusion priors and zero-shot adaptation to video, compared to their fine-tuning strategy.

**Video Denoising.** Video denoising, compared to VSR, is a simpler task for regression models, as they can often find the correct pixel value given a sufficiently large batch size. However, our method consistently outperforms others in terms of visual quality (LPIPS) and remains highly robust even as degradation becomes severe. Fig. 8 visualizes the denoising results on the REDS30 dataset.

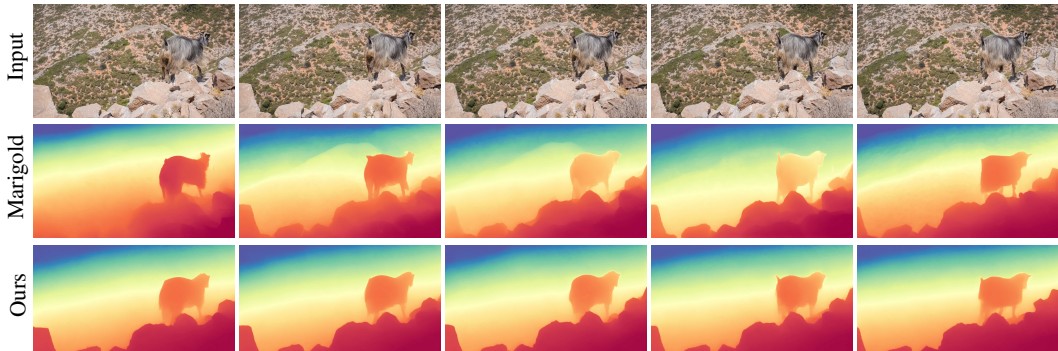

Figure 9: **Applying our techniques to consistent video depth.** Integrating our proposed framework into Marigold (Ke et al., 2024) helps improve the temporal consistency of video depth estimation.

Shift-Net (Li et al., 2023) fails to remove all noise, likely due to the out-of-domain noise level; VidToMe (Li et al., 2024) produces smooth results but lacks fine details. Although DiffBIR (Lin et al., 2024) generates highly detailed images, it suffers from poor temporal consistency, as evident in the changes to the pedestrian's head and the statue's face. In contrast, our method preserves both fine details and temporal consistency, effectively balancing these two aspects.

**Other Video Tasks: Consistent Video Depth.**   Our zero-shot framework is applicable to any pre-trained image-based diffusion models and could improve the predicted video consistency. Therefore, we integrate our proposed zero-shot framework into a state-of-the-art latent diffusion-based monocular depth estimator: Marigold (Ke et al., 2024). Fig. 9 shows that integrating our proposed framework into Marigold helps improve the temporal consistency of video depth estimation. We provide more visual comparisons in the supplementary materials.

This adaptability to various tasks (super-resolution, denoising, depth estimation) showcases the broad applicability of our approach. As more powerful or specialized image models emerge, our framework can quickly adapt to leverage these improvements for video restoration tasks. We provide computational complexity evaluations in the supplementary materials.

### 4.2 ABLATION STUDY

**Ways of Identifying Correspondence.**   Tab. 2 presents an ablation study comparing different approaches (optical flow and cosine similarity) for finding correspondences and their order in the UNet. As detailed in Sec. 3.3, the hybrid approach of using optical flow at the downsample blocks and cosine similarity at the upsample blocks achieves the best performance. Additionally, our proposed spatial-aware token merging further enhances performance by utilizing spatial information to guide correspondences. See supplementary materials for temporal profile comparisons.

**Applied Stages in the Denoising Process.**   Tab. 2 presents an ablation study evaluating the application of our two proposed components, hierarchical latent warping (HLW, Sec. 3.2) and hybrid spatial-aware token merging (HS-ToMe, Sec. 3.3), at the early, mid, and late stages of the denoising process. The results indicate that applying latent warping in the mid or late stages can significantly degrade the generated outcomes. Furthermore, ensuring consistency in the token space is crucial for achieving coherent and high-quality results.

### 5 CONCLUSION

We introduce a novel zero-shot video restoration framework utilizing pre-trained image-based diffusion models, eliminating the need for extensive retraining. Our approach integrates hierarchical latent warping and hybrid flow-guided, spatial-aware token merging, significantly enhancing temporal consistency and video quality under various degradation conditions. Experimental results demonstrate that our framework surpasses existing methods both in quality and consistency.

**Limitations.**   Our framework has two main limitations: (1) LDM decoder sensitivity can cause flickering in dynamic scenes. (2) Extreme degradation may yield unsatisfactory results. Future work will address these issues by stabilizing decoder output, and enhancing extreme degradation handling. Our framework's adaptability allows for the integration of future, more powerful diffusion models.

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

## A    APPENDIX / SUPPLEMENTAL MATERIAL

In this supplementary material, we first provide additional details on the testing datasets and evaluation metrics. Subsequently, we present more visual comparisons of various methods.

### A.1    ABLATION STUDIES ON CORRESPONDENCES IDENTIFIED BY COSINE SIMILARITY

Fig. 10 The figure shows the correspondences at denoising step 40 for three scenarios: without spatial awareness and padding removal, without spatial awareness, and with both spatial awareness and padding removal (ours). It is evident that padding values significantly affect the matching quality. However, even after removing padding, many mismatched diagonal lines remain, leading to blurry results. In contrast, our method effectively finds accurate correspondences by leveraging spatial information from the video.

### A.2    SEVERE DEGRADATION SCENARIOS.

Our balanced approach proves particularly effective in severe degradation scenarios. For instance, in $8\times$ super-resolution tasks, our method not only avoids artifacts but can even improve visual quality compared to per-frame approaches (Fig. 11). Additionally, in the $4\times$ video face super-resolution dataset (Chen et al., 2024), our results contain more details compared to FMA-Net and are temporally more consistent than per-frame method DiffBIR as shown in Fig. 14. This underscores the effectiveness of our ratio annealing technique in addressing the over-smoothing tendency while maintaining the benefits of our token merging approach. Additional comparisons on video super-resolution can be found at Fig. 12 and Fig. 13.

**Other Video Tasks: Consistent Video Depth.**    Our zero-shot framework is applicable to any pre-trained image-based diffusion models and could improve the predicted video consistency. Therefore, we integrate our proposed zero-shot framework into a state-of-the-art latent diffusion-based monocular depth estimator: Marigold (Ke et al., 2024). Fig. 15 shows that integrating our proposed framework into Marigold helps improve the temporal consistency of video depth estimation.

### A.3    COMPUTATIONAL COMPLEXITY

While our method focuses on zero-shot video restoration without additional training, it's important to consider the computational requirements in comparison to other approaches. Tab. 3 provides an overview of the training time and GPU specifications for different methods, including ours.

As shown in the table, our method stands out by not requiring any training or fine-tuning, which significantly reduces the computational resources needed. This is in stark contrast to other methods that require multiple high-end GPUs and several days of training time. For inference, our method introduces some computational overhead due to the hierarchical latent warping and hybrid token merging processes. However, this overhead is relatively small compared to the resources required for training or fine-tuning video models. Specifically, our method adds only approximately 6 seconds to the inference time of the base image diffusion model per frame.

### A.4    ADDITIONAL ABLATION STUDIES

**Comparison of temporal profiles.**    The comparisons in Fig. 16 also indicate that our results are smoother, demonstrating better temporal stability.

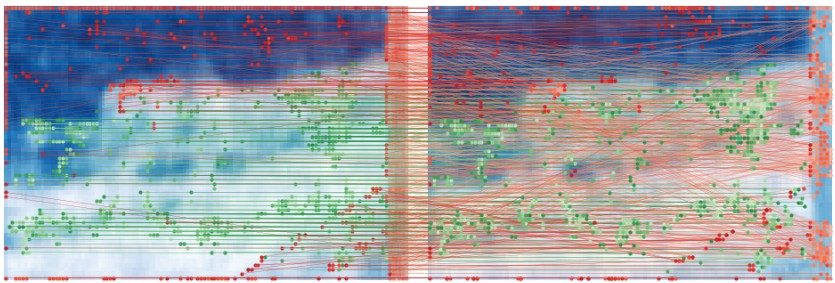

(a) Correspondences without spatial-awareness and padding removal

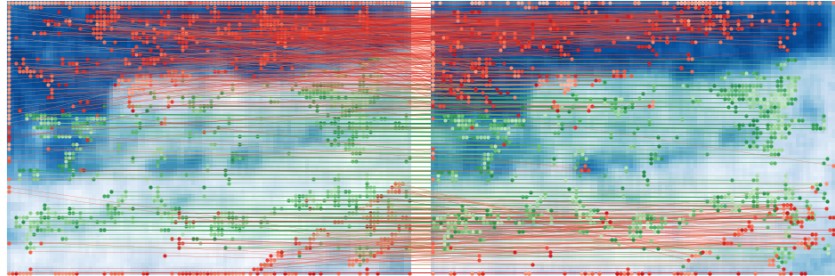

(b) Correspondences without spatial-awareness

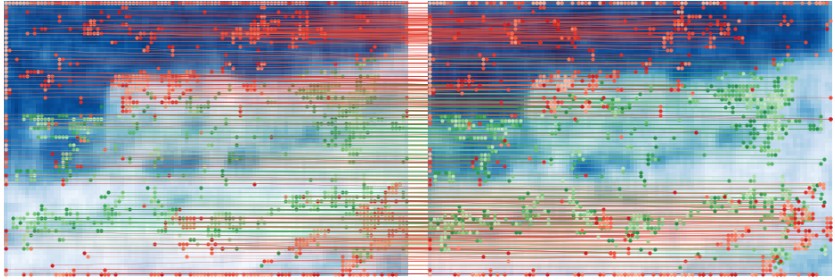

(c) Correspondences with spatial-awareness and padding removal (ours)

Figure 10: **Correspondences at denoising step 40 for different settings.**

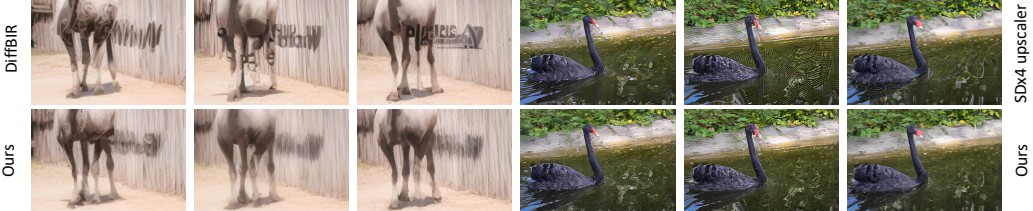

Figure 11: **Applying our method on DiffBIR and SD ×4 upscaler for 8×SR task.** In this case of severe degradation, our method avoids artifacts and outperforms per-frame inference in terms of visual quality.

**Token Unmerging Strategies.** We experimented with two unmerging strategies: averaging paired tokens and direct replacement with keyframe tokens. Tab. 4 shows the results of these experiments on the Vid4 x4 SR task. As shown in the table, the replacement method outperforms averaging in terms of LPIPS, indicating better perceptual quality. Our experiments consistently showed that averaging tends to produce blurrier outputs in restoration tasks. Based on these results, we adopted the replacement-based unmerging process in our final model, as it preserves more details and leads to sharper outputs.

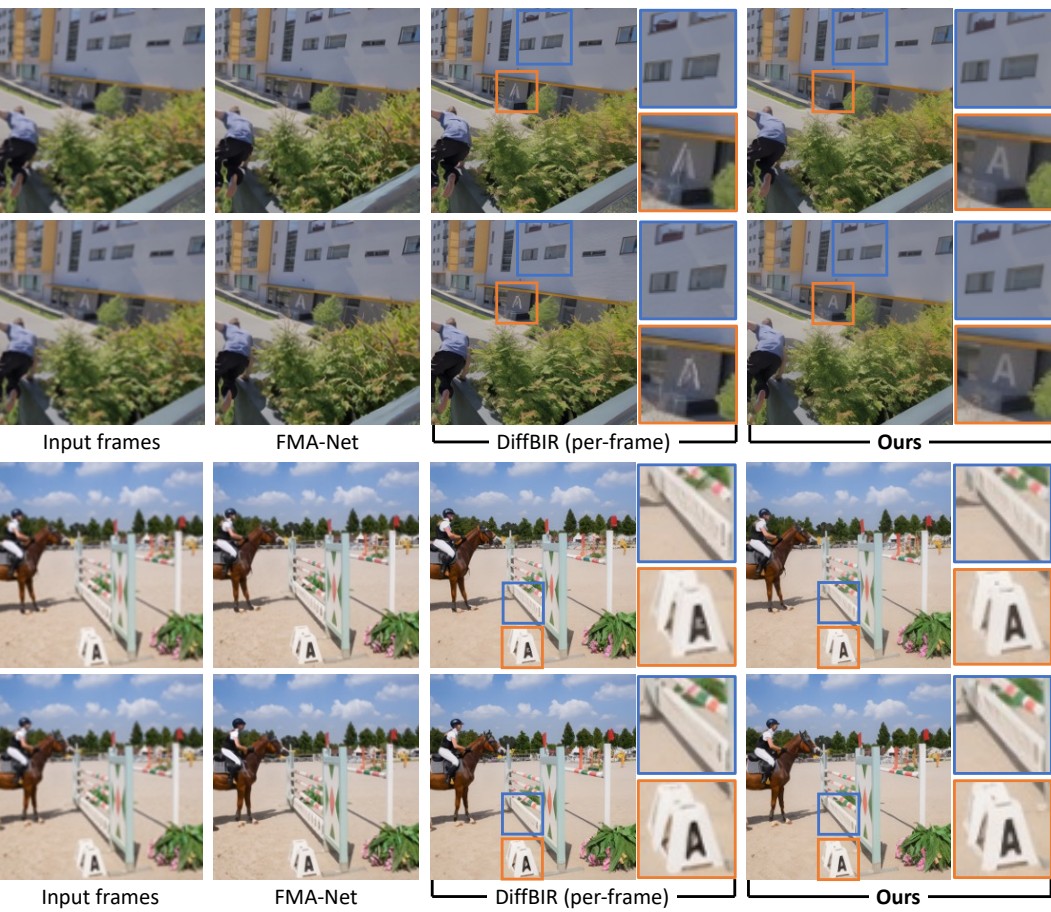

Figure 12: **Additional qualitative comparisons on 4× video super-resolution.** In the zoomed-in patches, our method produces clearer and more consistent results.

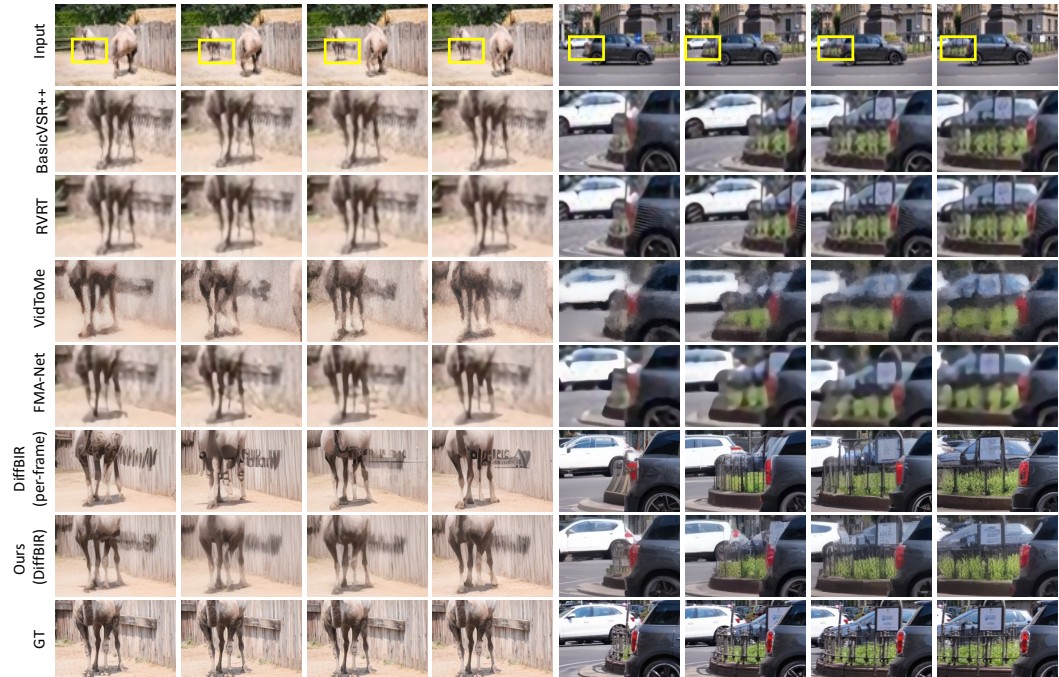

Figure 13: **Additional qualitative comparisons on 8× video super-resolution.** As shown in the first row, the low-quality input lacks almost all details. In the zoomed-in patches, our method produces clearer and more consistent results.

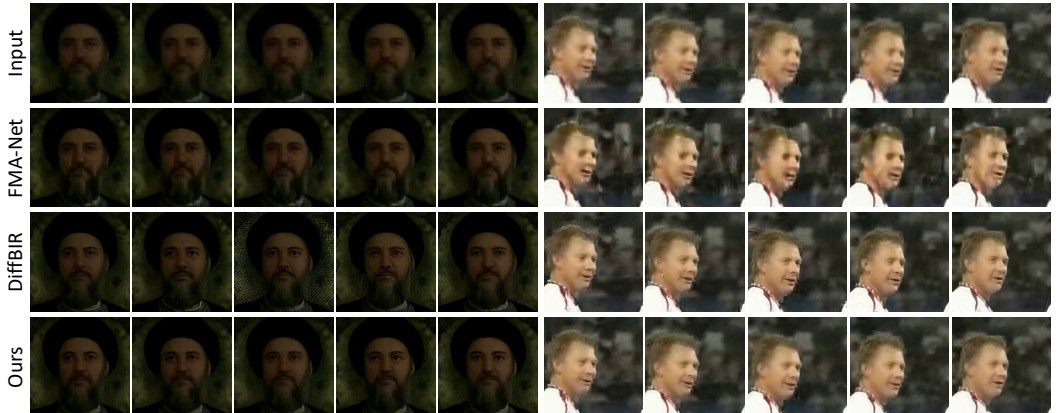

Figure 14: **Additional qualitative comparisons on 4× video face super-resolution.**

Table 3: **Training time and used devices for different methods.**

| Method | Training time | GPU specs |
|---|---|---|
| Shift-Net (Yan et al., 2018) | Not reported | 8 NVIDIA A100-32G GPUs |
| FMA-Net (Youk et al., 2024) | Not reported | Not reported |
| Upscale-A-Video (Zhou et al., 2023) | Not reported | 32 NVIDIA A100-80G GPUs |
| Ours | No training needed | - |

**Limitations: Extreme Degradation** Extreme degradation (*e.g.*, 32× super-resolution) or overly detailed facial features may yield unsatisfactory results (Fig. 17). However, our framework's adaptability allows the incorporation of future, more powerful image-based diffusion models. Future improvements will focus on refining keyframe selection, stabilizing decoder output across LDM ar-

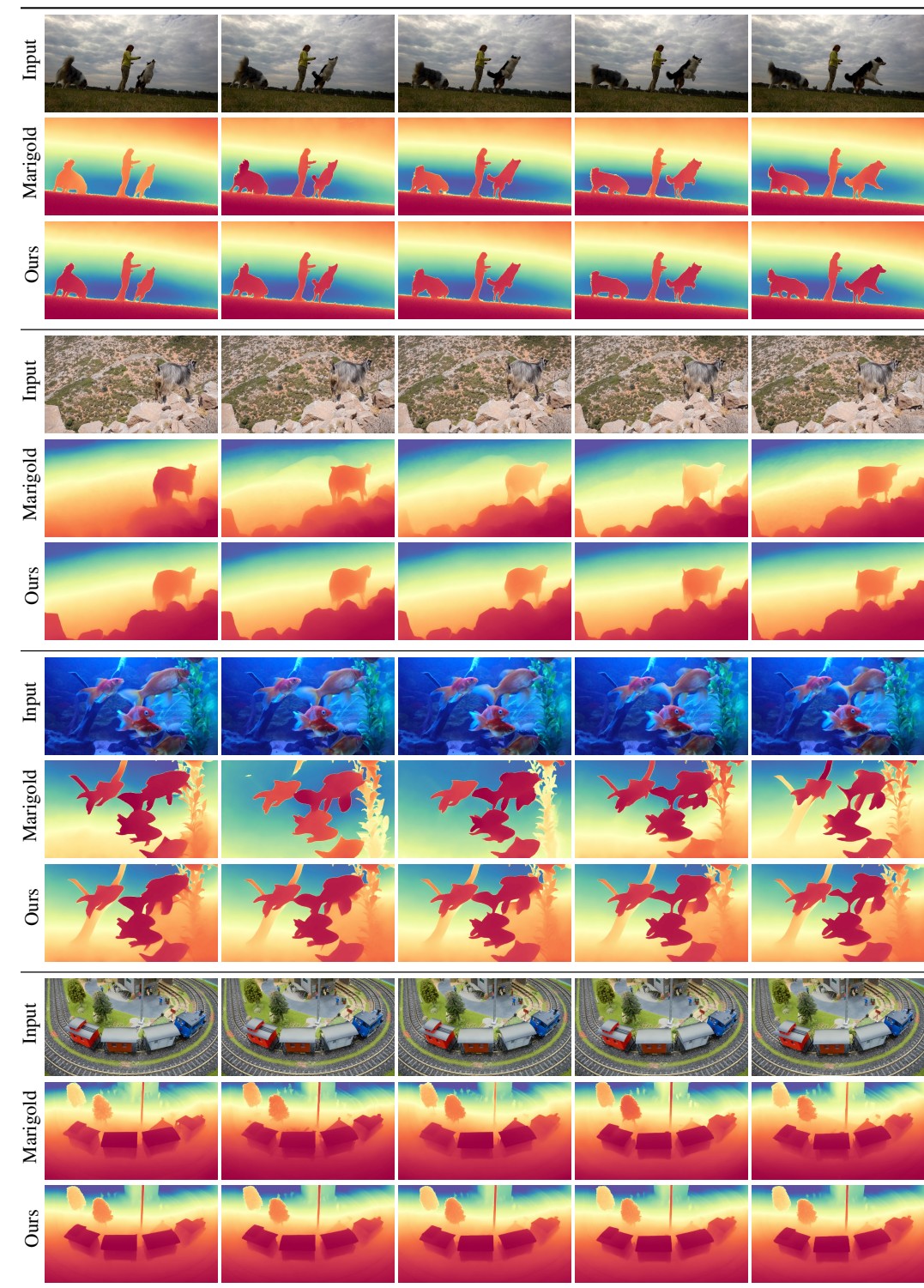

Figure 15: **Applying our techniques to consistent video depth.** Integrating our proposed framework into Marigold (Ke et al., 2024) helps improve the temporal consistency of video depth estimation.

chitectures, and enhancing extreme degradation handling. These aim to improve practical application and mitigate flickering issues inherent in LDM decoders.

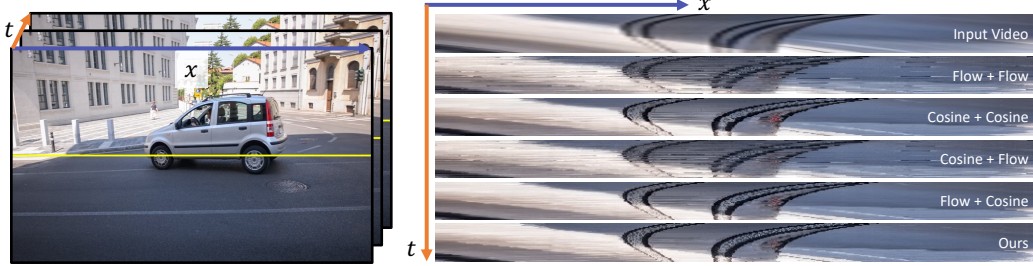

Figure 16: **Comparison of temporal profile.** We examine a row of pixels and track changes over time. The profiles from Flow + Flow and Cosine + Flow methods exhibit noise, indicating flickering artifacts. The Cosine + Cosine method shows smoother profiles but contains some discontinuities. Flow + Cosine demonstrates improved consistency but retains some distortions. Utilizing flow, cosine, and spatial-aware techniques, our method achieves the most seamless and consistent transitions, effectively minimizing artifacts.

Table 4: **Quantitative comparisons of different unmerging methods on Vid4 x4 SR task.**

| Unmerging Method | LPIPS $\downarrow$ |
|---|---|
| Averaging | 0.337 |
| Replacement | 0.329 |

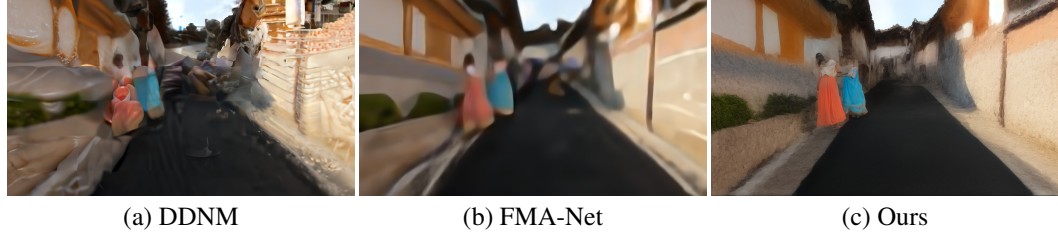

| (a) DDNM | (b) FMA-Net | (c) Ours |
|---|---|---|

Figure 17: **Failure case under 32x SR.** Most methods fail under this extreme degradation. However, if more powerful image-based diffusion models emerge in the future, our method can be easily adapted, offering greater potential to achieve this task.

