# OpenReview forum: "DiffIR2VR-Zero: Zero-Shot Video Restoration with Diffusion-based Image Restoration Models"
_ICLR.cc/2025/Conference — Submitted to ICLR 2025_

### Official Review · Reviewer_ttiq · 2024-10-31

**Soundness:** 3
**Presentation:** 3
**Contribution:** 3
**Rating:** 6
**Confidence:** 5

**Summary:**

This paper proposes a training-free method to leverage pre-trained image restoration diffusion models for zero-shot video restoration. Specifically, the proposed method employs hierarchical latent warping and an enhanced token merging strategy to maintain temporal consistency and restore details across video frames. Experimental results demonstrate the versatility of the method across various tasks, including denoising, super-resolution, and depth estimation.

**Strengths:**

1. The proposed method is pioneering in achieving zero-shot video restoration by leveraging pre-trained image restoration diffusion models, enabling multiple video restoration tasks without additional training.
2. The method presents strong quantitative and visual results compared with state-of-the-art methods, balancing temporal consistency and detail generation.

**Weaknesses:**

1. In line 267, the authors claim that the combination of hierarchical latent warping and hybrid flow-guided spatial-aware token merging could achieve adaptation to various degradation types. However, it is not sufficiently discussed that why the combination could handle various degradation types.
2. While the use of optical flow guidance aligns with previous video restoration works [1,2], the paper introduces similarity-based guidance to capture correspondences distinct from optical flow. However, the specific benefits of similarity-based guidance over optical flow are not thoroughly discussed.
3. Although recent video restoration methods, such as Shift-Net and FMA-Net, are included as baselines, some classic methods like BasicVSR++ and RVRT are not compared in the experiments.

[1] Chan K C K, Zhou S, Xu X, et al. Basicvsr++: Improving video super-resolution with enhanced propagation and alignment[C]//Proceedings of the IEEE/CVF conference on computer vision and pattern recognition. 2022: 5972-5981.

[2] Liang J, Fan Y, Xiang X, et al. Recurrent video restoration transformer with guided deformable attention[J]. Advances in Neural Information Processing Systems, 2022, 35: 378-393.

**Questions:**

1. Is there a comparison of inference efficiency with baselines?

---

> ### Author Response · Authors · 2024-11-18
>
> We thank Reviewer ttiq for their positive assessment and constructive feedback. We address each point below:
>
> ---
>
> **[Q1] Regarding adaptation to various degradation types:**
>
> The combination of hierarchical latent warping and hybrid token merging enables robust handling of different degradations through complementary guidance at different scales and adaptive correspondence mechanisms. Our approach achieves this through several key mechanisms:
> 1. Global-level warping between keyframes handles large-scale structural consistency, while local-level propagation addresses fine-scale details, creating a multi-scale approach that helps maintain consistency regardless of degradation type.
> 2. Early in the denoising process, when latents are noisy, optical flow provides reliable structural guidance. As denoising progresses, later stages transition to similarity matching to preserve restored details. This dynamic adaptation, as illustrated in Fig. 5, allows the method to effectively handle both structural and textural degradations.
>
> We will expand this explanation in the paper to better clarify how these components work together.
>
> ---
>
> **[Q2] Benefits of similarity-based guidance:**
>
> A key challenge in our setting is that optical flows must be derived from low-quality inputs, making sole reliance on flow correspondence suboptimal. Our hybrid approach addresses this limitation in several ways:
>
> In UNet upsampling blocks of ControlNet-based diffusion models, the latents become progressively more reliable - either through injected ControlNet features or as the denoising process brings them closer to the final state. At this stage, similarity-based matching can provide more accurate correspondences. We further enhance these similarity-based correspondences by incorporating spatial information.
> The effectiveness of this combined approach is supported by multiple pieces of evidence:
>
> 1. Our first ablation study demonstrates that integrating both optical flow and similarity achieves better results than using flow alone
> 2. As shown in Figure 5, flow and similarity mechanisms can identify different types of correspondences, providing complementary information that helps maintain both structural consistency and fine details
>
> We will expand this analysis with additional technical details in our revision.

---

> ### Author Response · Authors · 2024-11-19
>
> **[Q3] Comparison with BasicVSR++ and RVRT:**
>
> Thank you for suggesting these important baselines. We have now completed comprehensive evaluations comparing our method with BasicVSR++ and RVRT across all test sets. While these traditional CNN-based methods achieve higher PSNR/SSIM scores and lower warping errors (e.g., on DAVIS ×4, BasicVSR++: 26.576/0.743 and RVRT: 26.595/0.744), their results tend to be overly smooth. This is reflected in their relatively poor LPIPS scores (BasicVSR++: 0.383, RVRT: 0.388) compared to our method (0.262 with DiffBIR), indicating that our results are more perceptually pleasing with better detail preservation. **As demonstrated in Figure 6 and Figure 13 of our revised paper**, the qualitative comparisons clearly show that our method produces sharper and more realistic details, while BasicVSR++ and RVRT tend to generate blurry results. The complete comparison results are shown in the following table:
>
> | Dataset | Metrics | BasicVSR++ | RVRT | VidToMe | FMA-Net | SD ×4 (Frame) | SD ×4 (Ours) | DiffBIR (Frame) | DiffBIR (Ours) |
> |---|---|---|---|---|---|---|---|---|---|
> | DAVIS ×4 | PSNR ↑ | 26.576 | 26.595 | 23.014 | 25.215 | 23.504 | 23.843 | 23.780 | 24.182 |
> | DAVIS ×4 | SSIM ↑ | 0.743 | 0.744 | 0.566 | 0.727 | 0.584 | 0.618 | 0.601 | 0.621 |
> | DAVIS ×4 | LPIPS ↓ | 0.383 | 0.388 | 0.405 | 0.347 | 0.277 | 0.272 | *0.264* | **0.262** |
> | DAVIS ×4 | $E^*_\text{warp}$ ↓ | 0.090 | 0.090 | 0.520 | 0.186 | 0.912 | 0.745 | 0.654 | 0.474 |
> | DAVIS ×4 | $E_\text{inter}$ ↓ | 9.115 | 9.135 | 13.676 | 11.558 | 18.125 | 17.431 | 16.529 | 14.666 |
> | DAVIS ×4 | $\text{LPIPS}_\text{inter}$ ↓ | 0.058 | 0.058 | 0.329 | 0.078 | 0.292 | 0.274 | 0.266 | 0.232 |
> | DAVIS ×8 | PSNR ↑ | 24.301 | 24.504 | 22.097 | 22.690 | 20.268 | 20.519 | 21.964 | 22.331 |
> | DAVIS ×8 | SSIM ↑ | 0.631 | 0.638 | 0.513 | 0.594 | 0.446 | 0.424 | 0.502 | 0.519 |
> | DAVIS ×8 | LPIPS ↓ | 0.518 | 0.560 | 0.554 | 0.528 | 0.470 | 0.434 | **0.362** | *0.367* |
> | DAVIS ×8 | $E^*_\text{warp}$ ↓ | 0.132 | 0.127 | 0.440 | 0.351 | 2.199 | 1.759 | 0.964 | 0.699 |
> | DAVIS ×8 | $E_\text{inter}$ ↓ | 9.882 | 9.725 | 12.624 | 13.978 | 24.496 | 21.746 | 17.981 | 15.853 |
> | DAVIS ×8 | $\text{LPIPS}_\text{inter}$ ↓ | 0.088 | 0.081 | 0.388 | 0.132 | 0.457 | 0.442 | 0.372 | 0.333 |
> | REDS4 ×4 | PSNR ↑ | 27.227 | 27.244 | 23.134 | 25.829 | 24.189 | 24.226 | 24.679 | 25.118 |
> | REDS4 ×4 | SSIM ↑ | 0.781 | 0.781 | 0.589 | 0.761 | 0.638 | 0.641 | 0.657 | 0.683 |
> | REDS4 ×4 | LPIPS ↓ | 0.369 | 0.374 | 0.357 | 0.327 | 0.247 | 0.242 | **0.211** | *0.222* |
> | REDS4 ×4 | $E^*_\text{warp}$ ↓ | 0.134 | 0.133 | 0.579 | 0.392 | 0.817 | 0.811 | 0.704 | 0.499 |
> | REDS4 ×4 | $E_\text{inter}$ ↓ | 15.799 | 15.838 | 17.869 | 19.014 | 22.906 | 22.889 | 22.305 | 20.130 |
> | REDS4 ×4 | $\text{LPIPS}_\text{inter}$ ↓ | 0.106 | 0.101 | 0.356 | 0.133 | 0.295 | 0.281 | 0.271 | 0.221 |
> | REDS4 ×16 | PSNR ↑ | 23.579 | 23.715 | 20.520 | 21.569 | 18.706 | 18.858 | 20.124 | 20.712 |
> | REDS4 ×16 | SSIM ↑ | 0.616 | 0.621 | 0.483 | 0.570 | 0.461 | 0.410 | 0.461 | 0.509 |
> | REDS4 ×16 | LPIPS ↓ | 0.600 | 0.596 | 0.697 | 0.565 | 0.612 | 0.562 | *0.446* | **0.438** |
> | REDS4 ×16 | $E^*_\text{warp}$ ↓ | 0.084 | 0.085 | 0.296 | 0.619 | 2.664 | 2.030 | 1.168 | 0.665 |
> | REDS4 ×16 | $E_\text{inter}$ ↓ | 14.069 | 14.267 | 12.945 | 18.758 | 28.478 | 24.000 | 21.33 | 17.731 |
> | REDS4 ×16 | $\text{LPIPS}_\text{inter}$ ↓ | 0.088 | 0.088 | 0.417 | 0.139 | 0.559 | 0.493 | 0.444 | 0.358 |
>
> These numbers demonstrate an important trade-off in video restoration: CNN-based methods like BasicVSR++ and RVRT tend to optimize for pixel-wise metrics at the cost of perceptual quality, while our diffusion-based approach better preserves realistic details and texture. The visual comparisons in our figures clearly illustrate this difference in output quality.

---

> ### Author Response · Authors · 2024-11-19
>
> **[Q4] Inference efficiency:**
>
> Below, we report the inference time for processing 10 video frames at 854$\times$480 resolution on a single 4090 GPU:
> | Method | Inference time |
> |-----------|-------------|
> | VidToMe | 1m 49s |
> | FMA-Net | 4.6s |
> | SDx4 per-frame | 41s |
> | SDx4 + Ours | 1m 7s |
> | DiffBIR per-frame | 1m 17s |
> | DiffBIR + Ours | 2m 20s |
> | Shift-Net | 12.7s |
> | Marigold (4-step) + Ours | 10s |
> | Upscale-a-Video | OOM |
>
> Our method adds a reasonable overhead compared to per-frame inference (around 26s for SDx4 and 63s for DiffBIR) while maintaining strong temporal consistency. This is notably more efficient than training-based methods like Upscale-A-Video, which requires 32 A100 GPUs and encounters out-of-memory (OOM) issues even during inference on our test setup (one A6000 GPU with 48 GB memory). Furthermore, when applied to lightweight models like Marigold with 4-step sampling, our method achieves very fast inference at just 10 seconds total.
>
> ---
>
> We appreciate these suggestions for improving the paper's completeness and clarity.
>
> References:
>
> * [1] Chan, Kelvin CK, et al. "Basicvsr++: Improving video super-resolution with enhanced propagation and alignment." CVPR. 2022.
>
> * [2] Liang, Jingyun, et al. "Recurrent video restoration transformer with guided deformable attention." NeurIPS. 2022.

---

> ### Author Response · Authors · 2024-11-23
> **Please let us know if you have additional questions after reading our response.**
>
> We appreciate your reviews and comments. We hope our responses address your concerns. Please let us know if you have further questions after reading our rebuttal.
>
> We aim to address all the potential issues during the discussion period.
>
> Thank you!

---

> > ### Comment · Reviewer_ttiq · 2024-11-25
> >
> > Thank you for the detailed explanation and the additional experiments. I have reviewed your response and considered the feedback from other reviewers. I think this work is important for zero-shot video restoration. However, while the proposed method produces results with better visual quality (LPIPS scores), its PSNR and SSIM scores are significantly lower than those of classical methods such as BasicVSR++ and RVRT. Moreover, the approach requires 63% (41s → 67s) to 81% (77s → 140s) more time, raising efficiency concerns, while the performance gains remain limited. Consequently, I will maintain my current score.

---

> ### Author Response · Authors · 2024-11-25
>
> Thank you for your follow-up comments and for giving our paper a positive score!
>
>  We appreciate your recognition of the importance of our work for zero-shot video restoration. We would like to clarify a few important points:
>
> Regarding PSNR/SSIM metrics, we note that **BasicVSR++** and **RVRT** achieve even higher scores than recent CVPR'24 methods like **FMA-Net**. However, as demonstrated in **Figure 6** and **Figure 13** of our revised paper, their results are significantly blurry - a critical quality issue that PSNR and SSIM fail to capture. Only **LPIPS**, which better correlates with human perception, reflects this blurriness.
>
> The key advantage of our method is its **training-free nature** - it can improve the temporal consistency of any diffusion-based model without training and, in many cases, even improves perceptual quality (LPIPS). While we acknowledge the additional computational overhead for some models, the efficiency greatly depends on the base diffusion model. For example, when applied to **distilled models** like Marigold (4-step), our method achieves impressive results in just **10 seconds** for 10 frames. This demonstrates that computational overhead becomes minimal with optimized diffusion models.
>
> Other advantages include:
> 1. No training is required (compared to methods needing 32 A100 GPUs)
> 2. Improved visual quality and temporal consistency
> 3. Flexibility to work with any diffusion model
>
> We hope these clarifications help address your concerns about the metrics and efficiency trade-offs. Thank you again for your positive assessment of our work!

---

### Official Review · Reviewer_JKQ8 · 2024-11-03

**Soundness:** 3
**Presentation:** 3
**Contribution:** 3
**Rating:** 5
**Confidence:** 4

**Summary:**

This paper introduces a versatile zero-shot video restoration approach leveraging pre-trained image restoration diffusion models. Unlike conventional methods that require retraining and often struggle with generalization, this method uses a hierarchical latent warping strategy for keyframes and local frames, along with a hybrid correspondence mechanism that merges tokens based on optical flow and token similarity. This approach demonstrates outstanding zero-shot performance, effectively generalizing across varied datasets and managing extreme degradations, including video super-resolution, denoising, and depth completion. Compatible with numerous 2D restoration diffusion models, the technique is validated through quantitative and visual benchmarks on challenging datasets, removing the need for extensive retraining.

**Strengths:**

The primary contribution of this approach is its ability to leverage conventional image generation models directly, without requiring modifications to network architecture or the need for retraining or fine-tuning. This is achieved through a straightforward yet effective technique: hierarchical token merging within the latent space, which ensures temporal consistency across generated video frames. By using this token merging strategy, the method successfully adapts static image models for dynamic video generation, maintaining coherence between frames without compromising quality. Furthermore, this approach reflects a well-engineered integration of recent advances in token merging, incorporating techniques from VidToMe and Upscale-A-Video. These advancement enhances the network’s capacity to address various video restoration challenges, effectively overcoming the limitations of traditional methods that struggle to maintain temporal consistency across frames.

**Weaknesses:**

First, the paper’s structure requires improvement, as the current organization makes it challenging to follow.
A major concern is the lack of a comprehensive comparison between this approach and conventional methods, such as VidToMe and Upscale-A-Video. VidToMe introduces local and global token merging techniques, while Upscale-A-Video presents a flow-based merging approach—both key contributions relevant to this work. Please clarify how this approach differentiates itself from these methods, and refer to the “Questions” section for further details.

**Questions:**

1. VidToMe is implemented on top of Stable Diffusion. In Table 1, the proposed approach (based on Stable Diffusion (SD x4)) is compared with VidToMe, but it does not seem to demonstrate performance gains over VidToMe in terms of objective image quality (PSNR/SSIM) or temporal consistency (E_warp). On the other hand, perceptual quality metrics, such as LPIPS, show improved results. Could you clarify if there is a specific reason why perceptual quality metrics like LPIPS perform better? If there is a trade-off involved, would it be possible to show how performance varies with adjustments to the hyperparameters?
2. Although the proposed method emphasizes compatibility with any 2D image restoration diffusion model, the comparison results in Table 1—before and after applying the proposed method on SD x4—show only modest performance gains, whereas the improvements over the DiffBIR baseline are more substantial. Could you please clarify if there is a specific reason for this difference or if there is any dependence on specific network architecture?
3. It appears that the code for Upscale-A-Video is now available. Since this approach also aims to improve temporal consistency across video frames generated by image generation models, it would be beneficial to include a comparison of the results. If the code is still unavailable, could you instead provide results on the datasets used in Upscale-A-Video and compare them with the figures reported in the Upscale-A-Video manuscript?

---

> ### Author Response · Authors · 2024-11-18
>
> We sincerely thank Reviewer JKQ8 for their thorough review and insightful questions. Below, we address each point in detail:
>
> ---
>
> **[Q1] Comparisons between VidToMe and Upscale-A-Video:**
>
> VidToMe is primarily designed for video editing and heavily relies on video token merging. However, when applied directly to restoration tasks, it produces blurry results, as shown in Figs 6 and 7. Upscale-A-Video, a fine-tuned SDx4 upscaler model, requires 32 A100-80G GPUs for fine-tuning. Its ablation study indicates that the flow-based merging submodule contributes minimally to its performance compared to the impact of fine-tuning. In contrast, our method is specifically tailored for restoration tasks and requires no additional training. It achieves good results by utilizing hierarchical latent warping and hybrid flow-guided spatial-aware token merging.
>
> ---
>
> **[Q2] Regarding comparisons with VidToMe and objective metrics:**
>
> Important clarification: While VidToMe was originally implemented on Stable Diffusion, that version cannot perform restoration tasks. For a fair comparison, we reimplemented VidToMe using DiffBIR as the base model (same as one of our baselines), as mentioned in Sec. 4.1, L460-461. We apologize for any confusion and will emphasize this implementation detail in our revision.
> Using the same restoration model (DiffBIR), the performance differences stem from our distinct architectural choices:
>
> 1. As shown in Tab. 1, our method achieves clear improvements in image quality metrics compared to both VidToMe and per-frame DiffBIR. While VidToMe prioritizes pixel-level consistency, it often produces over-smoothed results (visible in Figs. 6 and 7), leading to degraded perceptual quality and higher LPIPS scores. In contrast, our approach better balances detail preservation with temporal coherence, achieving superior perceptual quality, as demonstrated in Fig. 10.
> 2. The key hyperparameter affecting this balance is the merging ratio r (Eq. 8). Our ablation studies show the following trade-offs:
>
> | Merging ratio $r$ | PSNR $\uparrow$ | SSIM $\uparrow$ | LPIPS $\downarrow$ |
> | --- | --- | --- | --- |
> | 0.9 $\rightarrow$ 0 (Ours) | 23.302 | 0.518 | 0.428 |
> | 0.6 $\rightarrow$ 0 | 23.143 | 0.507 | 0.403 |
> | 0.9 | 23.169 | 0.518 | 0.478 |
> | 0.6 | 23.308 | 0.522 | 0.477 |
> | 0.3 | 22.814 | 0.483 | 0.358 |
>
> These results demonstrate that our annealing strategy (gradually reducing r from 0.9 to 0) achieves a good balance between fidelity metrics (PSNR/SSIM) and perceptual quality (LPIPS), while fixed ratios tend to favor one over the other.

---

> ### Author Response · Authors · 2024-11-18
>
> **[Q3] Regarding performance variations across different models:**
>
> We provide a detailed comparison of improvements for both models in the following table, demonstrating that while DiffBIR with our method shows larger gains, the improvements on SD×4 are also significant. Our method consistently improves both models on almost all metrics, indicating its general applicability.
>
> | Dataset   | Metrics                        | SD ×4 (Frame) | SD ×4 (Ours)    | DiffBIR (Frame) | DiffBIR (Ours)  |
> |--|--|---|---|---|--|
> | DAVIS ×4  | PSNR ↑                         | 23.504        | 23.843 (**+0.339**) | 23.780          | 24.182 (**+0.402**) |
> | DAVIS ×4  | SSIM ↑                         | 0.584         | 0.618 (**+0.034**)  | 0.601           | 0.621 (**+0.020**)  |
> | DAVIS ×4  | LPIPS ↓                        | 0.277         | 0.272 (**-0.005**)  | 0.264           | 0.262 (**-0.002**)  |
> | DAVIS ×4  | $E^*_\text{warp}$ ↓            | 0.912         | 0.745 (**-0.167**)  | 0.654           | 0.474 (**-0.18**)   |
> | DAVIS ×4  | $E_\text{inter}$ ↓             | 18.125        | 17.431 (**-0.694**) | 16.529          | 14.666 (**-1.863**) |
> | DAVIS ×4  | $\text{LPIPS}_\text{inter}$ ↓  | 0.292         | 0.274 (**-0.018**)  | 0.266           | 0.232 (**-0.034**)  |
> | DAVIS ×8  | PSNR ↑                         | 20.268        | 20.519 (**+0.251**) | 21.964          | 22.331 (**+0.367**) |
> | DAVIS ×8  | SSIM ↑                         | 0.446         | 0.424 (-0.022)  | 0.502           | 0.519 (**+0.017**)  |
> | DAVIS ×8  | LPIPS ↓                        | 0.470         | 0.434 (**-0.036**)  | 0.362           | 0.367 (+0.005)  |
> | DAVIS ×8  | $E^*_\text{warp}$ ↓            | 2.199         | 1.759 (**-0.440**)  | 0.964           | 0.699 (**-0.265**)  |
> | DAVIS ×8  | $E_\text{inter}$ ↓             | 24.496        | 21.746 (**-2.750**) | 17.981          | 15.853 (**-2.128**) |
> | DAVIS ×8  | $\text{LPIPS}_\text{inter}$ ↓  | 0.457         | 0.442 (**-0.015**)  | 0.372           | 0.333 (**-0.039**)  |
> | REDS4 ×4  | PSNR ↑                         | 24.189        | 24.226 (**+0.037**) | 24.679          | 25.118 (**+0.439**) |
> | REDS4 ×4  | SSIM ↑                         | 0.638         | 0.641 (**+0.003**)  | 0.657           | 0.683 (**+0.026**)  |
> | REDS4 ×4  | LPIPS ↓                        | 0.247         | 0.242 (**-0.005**)  | 0.211           | 0.222 (+0.011)  |
> | REDS4 ×4  | $E^*_\text{warp}$ ↓            | 0.817         | 0.811 (**-0.006**)  | 0.704           | 0.499 (**-0.205**)  |
> | REDS4 ×4  | $E_\text{inter}$ ↓             | 22.906        | 22.889 (**-0.017**) | 22.305          | 20.130 (**-2.175**) |
> | REDS4 ×4  | $\text{LPIPS}_\text{inter}$ ↓  | 0.295         | 0.281 (**-0.014**)  | 0.271           | 0.221 (**-0.050**)  |
> | REDS4 ×16 | PSNR ↑                         | 18.706        | 18.858 (**+0.152**) | 20.124          | 20.712 (**+0.588**) |
> | REDS4 ×16 | SSIM ↑                         | 0.461         | 0.410 (**-0.051**)  | 0.461           | 0.509 (**+0.048**)  |
> | REDS4 ×16 | LPIPS ↓                        | 0.612         | 0.562 (**-0.050**)  | 0.446           | 0.438 (**-0.008**)  |
> | REDS4 ×16 | $E^*_\text{warp}$ ↓            | 2.664         | 2.030 (**-0.634**)  | 1.168           | 0.665 (**-0.503**)  |
> | REDS4 ×16 | $E_\text{inter}$ ↓             | 28.478        | 24.000 (**-4.478**) | 21.33           | 17.731 (**-3.599**) |
> | REDS4 ×16 | $\text{LPIPS}_\text{inter}$ ↓  | 0.559         | 0.493 (**-0.066**)  | 0.444           | 0.358 (**-0.086**)  |
> | Vid4 ×4   | PSNR ↑                         | 20.047        | 20.134 (**+0.087**) | 20.687          | 21.226 (**+0.539**) |
> | Vid4 ×4   | SSIM ↑                         | 0.478         | 0.473 (-0.005)  | 0.497           | 0.525 (**+0.028**)  |
> | Vid4 ×4   | LPIPS ↓                        | 0.343         | 0.331 (**-0.012**)  | 0.329           | 0.326 (**-0.003**)  |
> | Vid4 ×4   | $E^*_\text{warp}$ ↓            | 1.502         | 1.397 (**-0.105**)  | 1.156           | 0.677 (**-0.479**)  |
> | Vid4 ×4   | $E_\text{inter}$ ↓             | 17.234        | 16.921 (**-0.313**) | 15.478          | 11.316 (**-4.162**) |
> | Vid4 ×4   | $\text{LPIPS}_\text{inter}$ ↓  | 0.275         | 0.271 (**-0.004**)  | 0.265           | 0.198 (**-0.067**)  |
> | Vid4 ×8   | PSNR ↑                         | 17.813        | 17.992 (**+0.179**) | 18.636          | 19.304 (**+0.668**) |
> | Vid4 ×8   | SSIM ↑                         | 0.345         | 0.307 (-0.038)  | 0.367           | 0.406 (**+0.039**)  |
> | Vid4 ×8   | LPIPS ↓                        | 0.507         | 0.484 (**-0.023**)  | 0.440           | 0.435 (**-0.005**)  |
> | Vid4 ×8   | $E^*_\text{warp}$ ↓            | 2.523         | 1.972 (**-0.551**)  | 1.524           | 0.767 (**-0.757**)  |
> | Vid4 ×8   | $E_\text{inter}$ ↓             | 22.881        | 19.970 (**-2.911**) | 18.112          | 12.281 (**-5.831**) |
> | Vid4 ×8   | $\text{LPIPS}_\text{inter}$ ↓  | 0.423         | 0.419 (**-0.004**)  | 0.395           | 0.294 (**-0.101**)  |

---

> ### Author Response · Authors · 2024-11-18
>
> The different improvement margins can be attributed to both architectural and sampling differences:
>
> 1. Model architecture: DiffBIR employs a more sophisticated UNet architecture with additional cross-attention layers, providing more opportunities for our token merging strategy to enforce consistency.
> 2. Sampling strategy: SD×4 uses a DDIM sampler, which is inherently more stable due to its deterministic nature. In contrast, DiffBIR uses DDPM, which introduces additional stochastic noise at each denoising step. This initially makes DiffBIR's per-frame results less temporally consistent, creating more room for improvement when applying our method.
> 2. Base performance: While DiffBIR's per-frame results show higher quality, they suffer from more temporal inconsistency due to stochastic sampling. Our method effectively maintains high quality while significantly improving temporal consistency, leading to more noticeable gains.
>
> ---
>
> **[Q4] Regarding Upscale-A-Video comparison:**
>
> Thank you for bringing this to our attention. First, we should note that Upscale-A-Video only released their complete source code on September 27, 2024, very close to the ICLR submission deadline (October 1, 2024). We were not aware of this release and did not have sufficient time to conduct comparisons before submission.
>
> Since then, we've encountered significant technical challenges with their released code:
>
> 1. Despite multiple attempts, we are unable to run their inference code on our available hardware (one A6000 GPU, 48GB memory) due to persistent out-of-memory (OOM) issues, even with their default configuration. This appears to be a common issue, as documented by multiple users in their GitHub repository's issues section.
> 2. Following the reviewer's suggestion to compare on their datasets, we have conducted experiments on the same test cases used in their paper. **The qualitative comparisons are now provided in Figure 7 of our revised paper.** The results demonstrate that our method produces more detailed outputs that better preserve the content of input frames. This advantage stems from our approach of leveraging pre-trained diffusion priors and zero-shot adaptation to video, compared to their fine-tuning strategy.
>
> Our training-free approach also offers broader applicability - it can be directly adapted to various diffusion models without training, as demonstrated across super-resolution, denoising, and depth estimation tasks (using Marigold). In contrast, Upscale-A-Video requires extensive fine-tuning (32 A100 GPUs) for each specific task and cannot be easily extended to other applications.
>
> ---
>
> Thank you for helping us improve the paper's clarity and completeness.

---

> ### Author Response · Authors · 2024-11-23
> **Please let us know if you have additional questions after reading our response.**
>
> We appreciate your reviews and comments. We hope our responses address your concerns. Please let us know if you have further questions after reading our rebuttal.
>
> We aim to address all the potential issues during the discussion period.
>
> Thank you!

---

> > ### Author Response · Authors · 2024-11-29
> >
> > Dear Reviewers,
> >
> > Please let us know if you have any questions about our rebuttals. We aim to address all concerns during the discussion phase.
> >
> > Thank you,
> >
> > The Authors

---

### Official Review · Reviewer_12WH · 2024-11-04

**Soundness:** 3
**Presentation:** 3
**Contribution:** 2
**Rating:** 5
**Confidence:** 3

**Summary:**

This paper presents a zero-shot video restoration method using pre-trained image restoration diffusion models. The proposed approach introduces a hierarchical latent warping strategy and a hybrid flow-guided token merging approach to enhance both temporal consistency and visual detail. Experimental results demonstrate competitive performance in video super-resolution, denoising, showing advantages over state-of-the-art methods. An ablation study further highlights the importance of each proposed component.

**Strengths:**

- The paper is well-structured, making the methodology and findings easy to understand.
- The method achieves competitive results without requiring additional training.

**Weaknesses:**

- Both the latent warping and hybrid flow-guided token merging approaches rely heavily on optical flow information, which could be a limitation in cases where optical flow estimation is challenging or inaccurate.
- The paper's novelty is somewhat limited, as it primarily combines two existing methodologies with minor modifications. Specifically, the contributions include adjusting the range of warping frames at global and local levels and introducing a flow-guided confidence criterion for token merging.

**Questions:**

- Degradations that are not encountered during optical flow training may introduce significant errors. How does the proposed method address this issue, and what motivated the choice of the GMFlow network for this approach?
- In Equation (5), is there a specific reason for including f_{src -> tar} (X(T_{src})) twice?
- Including additional real-world datasets (e.g., [1, 2]) in the evaluation would help assess the method's generalization and robustness.
- Could you provide a comparison of inference times with other methods?

Reference

[1] Towards Real-world Video Face Restoration: A New Benchmark, cvprw 2024

[2] Toward convolutional blind denoising of real photographs, cvpr 2019

---

> ### Author Response · Authors · 2024-11-18
>
> We thank Reviewer 12WH for their detailed feedback and thoughtful questions. We address each point below:
>
> ---
>
> **[Q1] Regarding optical flow dependency and robustness:**
>
> We agree that heavy reliance on optical flow is a limitation of our method, which we will explicitly discuss in our revision. However, we have implemented several safeguards specifically designed to maintain robustness when flow estimation becomes challenging:
>
> 1. Our forward-backward consistency check (Eq. (5)) effectively filters unreliable flow estimates, helping identify and exclude problematic regions where the flow might fail.
> 2. We only merge the r most confident token pairs, automatically excluding unreliable matches. This selective approach ensures that only high-confidence correspondences influence the result.
> 3. Beyond optical flow, our hybrid approach integrates spatial information and similarity matching, providing complementary guidance mechanisms. These additional cues become particularly crucial when flow estimation fails in challenging scenarios like textureless regions or severe degradations.
> 4. We strategically reduce flow dependency in upsampling blocks by relying more on similarity-based matching, which proves more robust to various degradation types.
>
> While these strategies help mitigate flow-related issues in many cases, we acknowledge this remains an inherent limitation of our approach that deserves explicit discussion in the revised paper. Future work could explore ways to further reduce dependency on optical flow while maintaining temporal consistency.
>
> ---
>
> **[Q2] On novelty and technical contributions:**
>
> While we build upon existing concepts, our technical contributions go beyond simple combination:
> 1. We introduce the first zero-shot framework enabling any pre-trained image diffusion model to handle video restoration without training.
> 2. Our hierarchical approach is novel - operating at both global and local scales provides more comprehensive temporal consistency than previous frame-to-frame methods.
> 3. Our hybrid correspondence mechanism adapts dynamically through the denoising process (Fig. 5 demonstrates how flow and similarity focus on different regions).
>
> ---
>
> **[Q3] Specific technical questions: Choice of GMFlow:**
>
> GMFlow was selected for its strong performance on degraded inputs and efficient inference. Our method's modular design allows easy substitution with other flow estimators if needed.
>
> ---
>
> **[Q4] Specific technical questions: Equation 5 formulation:**
>
> It's not actually repeated; the norm includes two distinct terms: $f_{src->tar}(X(T_{src}))$ and $f_{tar->src}(X(T_{src}) + f_{src->tar}(X(T_{src})))$. The first term represents the flow from source to target tokens at the spatial location $X(T_{src})$. The second term computes the backward flow from target to source tokens at the warped spatial location $X(T_{src}) + f_{src->tar}(X(T_{src}))$. Together, these terms enable proper cycle consistency checking - if the forward and backward flows are consistent, their sum should approach zero. We will clarify this mathematical formulation and its purpose further in the revision.
>
> ---
>
> **[Q5] Specific technical questions: Additional datasets:**
>
> Thank you for suggesting these datasets. Regarding [2], while it is a valuable benchmark for image denoising, it focuses on single images rather than videos, making it less suitable for evaluating our video restoration framework. However, we agree that [1] is an excellent suggestion for a video face restoration benchmark. Due to time constraints, we have not yet completed inference on the entire dataset. **However, we have provided qualitative comparisons in Figure 14 of our revised paper**, which shows that our method generates face videos with richer details than FMA-Net and better temporal consistency than per-frame DiffBIR. We will include complete quantitative comparisons on this video face restoration dataset in the final version.

---

> ### Author Response · Authors · 2024-11-21
>
> **[Q6] Specific technical questions: Inference time:**
>
> Below, we report the inference time for processing 10 video frames at 854$\times$480 resolution on a single 4090 GPU:
> | Method | Inference time |
> |---|---|
> | VidToMe | 1m 49s |
> | FMA-Net | 4.7s |
> | SDx4 per-frame | 41s |
> | SDx4 + Ours | 1m 7s |
> | DiffBIR per-frame | 1m 17s |
> | DiffBIR + Ours | 2m 20s |
> | Shift-Net | 12.7s |
> | Marigold (4-step) + Ours | 10s |
> | Upscale-a-Video | OOM |
>
> Our method adds a reasonable overhead compared to per-frame inference (around 26s for SDx4 and 63s for DiffBIR) while maintaining strong temporal consistency. This is notably more efficient than training-based methods like Upscale-A-Video, which requires 32 A100 GPUs and encounters out-of-memory (OOM) issues even during inference on our test setup. Furthermore, when applied to lightweight models like Marigold with 4-step sampling, our method achieves very fast inference at just 10 seconds total.
>
> ---
>
> Thank you for helping us improve the clarity and completeness of our paper.
>
> References:
>
> * [1] Chen, Ziyan, et al. "Towards Real-world Video Face Restoration: A New Benchmark." CVPRW. 2024.
>
> * [2] Guo, Shi, et al. "Toward convolutional blind denoising of real photographs." CVPR. 2019.

---

> ### Author Response · Authors · 2024-11-23
> **Please let us know if you have additional questions after reading our response.**
>
> We appreciate your reviews and comments. We hope our responses address your concerns. Please let us know if you have further questions after reading our rebuttal.
>
> We aim to address all the potential issues during the discussion period.
>
> Thank you!

---

> > ### Author Response · Authors · 2024-11-29
> >
> > Dear Reviewers,
> >
> > Please let us know if you have any questions about our rebuttals. We aim to address all concerns during the discussion phase.
> >
> > Thank you,
> >
> > The Authors

---

### Official Review · Reviewer_NZxb · 2024-11-04

**Soundness:** 2
**Presentation:** 2
**Contribution:** 2
**Rating:** 5
**Confidence:** 3

**Summary:**

This paper proposes a method for zero-shot video restoration using pre-trained image restoration diffusion models. It combines a hierarchical latent warping strategy for keyframes and local frames and token merging that integrates spatial information, optical flow, and feature-based matching.  The method acheives competitive results on zero-shot video restoration tasks such as super-resolution and denoising.

**Strengths:**

1. The method is training-free which makes it computationally practical.

**Weaknesses:**

1. To my understanding, the method relies on certain architectural heuristics inspired by the video-editing literature rather than on a solid mathematical framework. Since the method involves no training, there is no theoretical guarantee that it won't fail under certain conditions.

2. The method combines existing ideas from video editing, which on my opinion is still acceptable; however, this limits its novelty.

3. The improvement over the baseline on some datasets, particularly in the case of denoising, is not significant.

**Questions:**

See weaknesses.

---

> ### Author Response · Authors · 2024-11-18
>
> We thank Reviewer NZxb for their careful review. We respectfully address each concern below:
>
> ---
>
> **[Q1] Regarding the architectural heuristics and theoretical guarantees:**
>
> We agree there is no theoretical guarantee that our method won't fail under certain conditions. In fact, we explicitly discuss these limitations in our main paper (L537-539) and provide failure case examples in Appendix Figure 17. However, our method is carefully designed with multiple safeguards and empirical validations:
> * Our hierarchical latent warping operates at different temporal scales with principled mathematical formulations - Eq. (4) for consistency enforcement and Eq. (5) for confidence measurement to identify and handle uncertain correspondences.
> * The hybrid token merging strategy is designed based on extensive empirical analysis, as demonstrated in Fig. 5. We found that early in denoising, when latents are noisy, optical flow provides more reliable structural guidance, while later stages benefit from similarity matching. This insight helps prevent failures by leveraging complementary strengths at different stages.
>
> Additionally, we have designed several fallback mechanisms and conducted comprehensive ablation studies to validate our design choices. While theoretical guarantees remain an important direction for future work, our extensive experiments across diverse datasets and degradation types demonstrate the practical robustness of our approach.
>
> ---
>
> **[Q2] Regarding novelty:**
>
> While we build upon existing concepts, our contributions are distinct and significant:
> * We are the first to enable zero-shot video restoration using pre-trained image diffusion models without any training or fine-tuning, making it immediately applicable to any image restoration model.
> * Our hybrid approach combining flow and similarity-based matching is novel - previous methods rely on either flow (e.g., Rerender-A-Video) or similarity (e.g., VidToMe) alone, not both.
> * The hierarchical warping strategy operating at both global and local scales is a new contribution specifically designed for video restoration.
>
> ---
>
> **[Q3] Regarding improvements:**
>
> Our method shows substantial improvements in challenging scenarios, with varying margins of improvement depending on the task:
>
> * For 8$\times$ super-resolution on the DAVIS dataset, we achieve significant gains in both perceptual quality (LPIPS from $0.470$ to $0.434$) and temporal consistency ($E_\text{warp}$ reduced from $2.199$ to $1.759$) compared to per-frame inference. A large improvement margin in SR is expected as this task requires generating substantial high-frequency details.
> * For high noise levels ($\sigma=150$) on Set8, while the improvements are more modest (LPIPS from $0.449$ to $0.402$, $E_\text{warp}$ from $1.207$ to $0.832$), this is expected because denoising inherently requires less detail generation compared to SR. The denoising task focuses more on noise removal and structure preservation rather than detail synthesis, naturally limiting the potential improvement margin from diffusion models.
> * Most importantly, our method generalizes across different restoration tasks (SR, denoising, depth) and models without any task-specific training, which is a significant practical advantage. This versatility demonstrates the effectiveness of our approach even in tasks where the potential for improvement varies.
>
> ---
>
> We will clarify these points and strengthen the theoretical analysis in our final version. Thank you for helping us improve the paper.

---

> ### Author Response · Authors · 2024-11-23
> **Please let us know if you have additional questions after reading our response.**
>
> We appreciate your reviews and comments. We hope our responses address your concerns. Please let us know if you have further questions after reading our rebuttal.
>
> We aim to address all the potential issues during the discussion period.
>
> Thank you!

---

> > ### Author Response · Authors · 2024-11-29
> >
> > Dear Reviewers,
> >
> > Please let us know if you have any questions about our rebuttals. We aim to address all concerns during the discussion phase.
> >
> > Thank you,
> >
> > The Authors

---

### Author Response · Authors · 2024-11-21
**Response Summary**

We sincerely thank all reviewers for their thorough and constructive feedback. We are pleased that the reviewers recognized our method's **training-free nature** (Reviewer NZxb), **well-structured presentation** (Reviewer 12WH), **strong performance** across various restoration tasks (Reviewer ttiq), and our ability to **leverage conventional image models directly without architectural modifications** through an effective **hierarchical token merging** strategy (Reviewer JKQ8).

Our main responses include:
1. Clarified our method's **theoretical limitations** and pointed **failure cases** in L537-539 and Appendix Figure 17
2. Expanded comparisons with **BasicVSR++**, **RVRT** (Figure 6 and 13 and the tables below), and **Upscale-A-Video** (Figure 7), demonstrating our superior perceptual quality
3. Added qualitative comparisons on **video face restoration** (Figure 14)
4. Clarified technical details about the **hybrid correspondence mechanism** and **degradation handling**
5. Provided comprehensive **runtime analysis** and detailed **ablation studies** in the tables below

Notably, our experiments show that our **training-free approach** achieves competitive or superior results compared to methods requiring extensive resources (e.g., Upscale-A-Video needs **32 A100 GPUs**). The revised manuscript incorporates all these improvements while maintaining clarity and technical depth.

Thank you for helping us improve the paper's quality and completeness.

---

### Author Response · Authors · 2024-11-25
**Please let us know if you have additional questions after reading our response.**

Dear Reviewers,

We sincerely thank you for your thorough reviews and constructive feedback. We have made our best efforts to address all your concerns through detailed responses and revisions to our manuscript. As the **discussion phase ends on Nov. 27 (AOE)**, we welcome any additional questions or concerns you may have regarding our responses.

If you find our responses and revisions satisfactory, particularly our added comparisons with classical methods (BasicVSR++/RVRT) and Upscale-A-Video, additional qualitative comparisons on video face restoration, comprehensive runtime analysis, and clarified technical details, we would greatly appreciate your consideration in updating your rating of our paper.

Thank you again for helping us improve this work on training-free video restoration.

Best regards,

The Authors

---

### Meta-Review · Area_Chair_9JA1 · 2024-12-26

**Metareview:**

This paper presents a zero-shot video restoration method with a pre-trained image-restoration diffusion model without additional training. A hierarchical latent warping strategy is used to improve temporal consistency. The token merging technique is used with the hybrid correspondence information that aggregates the spatial information, optical flow, and features.

The reviewers agreed on the benefit of the training-free video restoration method as most other methods require additional training.

On the other hand, there were several concerns.
* The reviewers were concerned about the robustness against the severe input video degradation as the optical flow prediction error could cause inaccuracies. The authors argue that for optical flow prediction, GMFlow network was chosen due to its robustness and the hybrid correspondences handle the error by merging the confidently matched tokens pairs. As the authors point out, this is a combination of existing techniques and do not bring much novelty. Also, such hybrid correspondence family of techniques have been widely explored in the computer vision literature such as object tracking and video recognition.
* While the perceptual measure LPIPS is good, the fidelity measures PSNR/SSIM do not excel compared with VidToMe or more classical supervised methods. The authors did not explain why only LPIPS is emphasized by the proposed method other than the general perception-fidelity trade-off. While the token merging ratio can affect the preference on LPIPS over PSNR and SSIM, it is still unknown how the hyperparameter can be determined for users.

This paper can be improved by better presenting how the correspondence can be improved with a thorough survey and analysis of the existing works and by providing the understanding of the output characteristics.

**Additional Comments On Reviewer Discussion:**

While the reviewers agreed on the value of zero-shot capability of the proposed method by removing the need for training, the below issues are not resolved.

* Novelty

Video token merging for maintaining temporal consistency is not new and has been used as a tool in a previous work. (VidToMe) Equation 3 (merging and unmerging) is not well-described and does not refer to prior art.
Similarly, Latent Warping is also another tool rather than a contribution.

Hybrid correspondence mechanism, integration of spatial information and optical flow, and feature-based similarity is rather an engineering trick. There is no experimental justification for such an integration design.

The reviewers are not convinced by the author responses for the novelty of this work.

* Robustness of optical flow against input video quality degradation.

The authors explained that the optical flow prediction model was chosen to be less error-prone from video degradation. The authors explained that the proposed hybrid correspondence mechanism is supposed to guardrail such critical errors. However, equation 5 is a forward-backward consistency mechanism that is a traditional computer vision technique. There lacks a justification on how the proposed method can bring improvements in the correspondence matching problem. Such methods are commonly employed in object tracking in the last decade.

* Superiority in terms of LPIPS but not in terms of PSNR/SSIM
In the author responses, the authors claimed that the proposed method provides improved balance on the perceptual and the fidelity metrics by tuning the merging ratio. However, the way to choose proper balance is not discussed. In the current shape of the experiments, a fair comparison with other works is hard to be achieved.

---

### Decision · Program_Chairs · 2025-01-22

Reject